

# Assimilating Multi-site Eddy-Covariance Data to Calibrate the CH$_4$ Wetland Emission Module in a Terrestrial Ecosystem Model

Jalisha Theanutti Kallingal[1], Marko Scholze[1], Paul Anthony Miller[1], Johan Lindström[2], Janne Rinne[3], Mika Aurela[4], Patrik Vestin[1], and Per Weslien[5]

[1]Department of Physical Geography and Ecosystem Science, Lund University, Lund, Sweden, S-223 62
[2]Centre for Mathematical Sciences, Lund University, Lund, Sweden, S-223 62
[3]Natural Resources Institute Finland, Helsinki, Finland, FI-00 790
[4]Finnish Meteorological Institute, Helsinki, Finland, FI-00101
[5]Department of Geosciences, University of Gothenburg, Sweden, S-413 90

**Correspondence:** Jalisha T. Kallingal (jalisha.theanutti@nateko.lu.se)

**Abstract.** In this study, we use a data assimilation framework based on the Adaptive Markov Chain Monte Carlo (MCMC) algorithm to constrain process parameters in LPJ-GUESS model using CH$_4$ eddy covariance flux observations from 14 different natural boreal, temperate and arctic wetlands. The objective is to derive a single set of calibrated parameter values. The calibrated parameter values are then used in the model to validate its CH$_4$ flux output against independent CH$_4$ flux observa-

tions from five different types of natural wetlands situated in different locations, assessing their generality for simulating CH$_4$ fluxes from boreal, temperate and arctic wetlands. The results show that the MCMC framework has substantially reduced the cost function (measuring the misfit between simulated and observed CH$_4$ fluxes) and facilitated detailed characterisation of the posterior parameter distribution. A reduction of around 50% in RMSE compared to the observations was achieved after optimisation. The results of validation experiment indicate that for four out of the five validation sites the RMSE was successfully

reduced, demonstrating the effectiveness of the framework for estimating CH$_4$ emissions from wetlands not included in the assimilation experiment. For wetlands above 45°N, the total mean annual CH$_4$ emission estimation using the optimised model resulted in 28.16 Tg C y$^{-1}$, and for regions above 60 °N, it resulted in 7.46 Tg C y$^{-1}$.

## 1   Introduction

Methane (CH$_4$) emissions from wetlands contribute 20-30 % to the total global emissions (IPCC AR6 chapter 5: Canadell et al.

(2022), Saunois et al. (2020)). About one-third to one-half of these wetland emissions are from wetlands located at northern latitudes of North America, Europe and Russia (Saunois et al., 2016a). According to the IPCC AR6 report, wetlands are the largest single source of uncertainty to the global CH$_4$ budget estimate. It is expected to have increased uncertainties in wetland CH$_4$ emissions in the future (Christensen et al., 2007), partly due to climate change and partly due to spatio-temporal changes in wetland extent (that in itself is partly a consequence of climate change) (Saunois et al. (2016b), Zhang et al. (2017)). A

key question to consider here is the extent to which these changes in emissions are occurring and how they will impact the future global greenhouse gas (GHG) budget and hence the climate. While current in-situ measurement techniques such as



eddy-covariance (EC) flux observations are promising for drawing assumptions on this issue at local scales, studies to date have faced difficulties in estimating wetland $CH_4$ emissions over large landscapes (Saunois et al., 2020).

An attempt to overcome this limitation through process-based modelling of global $CH_4$ emissions was first initiated by Fung
et al. (1991) followed by Christensen and Cox (1995) and more mechanistically by Cao et al. (1996), and Walter and Heimann (2000). These models were simple in structure, and later more attention was given to model process improvement through the studies of mainly Segers and Leffelaar (2001), Gedney et al. (2004), and Zhuang et al. (2006). In the last decade more detailed models with more complexity and a wider range of applications were developed by Wania et al. (2010), Ringeval et al. (2010), Susiluoto et al. (2018) etc. All of these past efforts indicate that comprehensive, process-based modelling of $CH_4$ emissions
from wetland ecosystems is unquestionably a key way to understand the variability of wetlands and how they respond to environmental stresses and climate change (Saunois et al., 2020).

As all these models are approximations of the real world and exhibit their own uncertainties, here again the question is how to reduce the uncertainty for large-scale applications. According to Kuppel et al. (2012) every terrestrial biosphere model contains uncertainties in five different ways: errors in real data used for calibration, errors in meteorological forcing, errors in
process descriptions, errors in model parameter values, and inaccurate initial state of the model. The first two errors are related to measurement, while the last three are related to model formulation and are important to improve the model performance for general applications. There has been a growing effort to reduce uncertainty related to the last three sources of error factors in several ways. A popular method to reduce uncertainty in model parameters is to calibrate the model simulations against observations. Previous studies like Williams et al. (2009), Susiluoto et al. (2018) and Kuppel et al. (2012), based on different
models, different data and different parameter sets provide examples of improving model parameters and reducing uncertainties through data assimilation.

In this study, we consider uncertainties in parameter values of the $CH_4$ module of a global process-based ecosystem model, Lund–Potsdam–Jena General Ecosystem Simulator (LPJ-GUESS) v4.1 and aim to reduce their uncertainties. Dynamic Global Vegetation Models (DGVMs) like LPJ-GUESS are state-of-the-art tools for studying the functioning of high-latitude wetlands
and estimating the dynamics in their global carbon balance (Sitch et al., 2003). In a previous study, Kallingal et al. (2023) have used EC flux observations collected at an individual site to investigate the potential of a Markov Chain Monte Carlo (MCMC) type (Global Rao-Black-wellised Adaptive Metropolis, GRaB-AM) algorithm to optimise the parameters in the $CH_4$ module of LPJ-GUESS. The study showed that EC flux measurements of $CH_4$ contains useful information for optimising the $CH_4$ model parameters due to the high temporal resolution of the $CH_4$ flux measurements. However, the small spatial
scale (site scale) and limited temporal extent of data collected from a single site could have over fitted parameters to the specificities of the particular site used. This points to the need for a more general approach. For example, in a study conducted by Groenendijk et al. (2011) the parameters of a photosynthesis model are optimised using EC flux observations from several Fluxnet sites. Similarly, studies like Kuppel et al. (2012) and Raoult et al. (2016) have constrained the parameters of a global ecosystem model using multi-site EC flux observations. Considering these studies and the results of Kallingal et al. (2023),
we hypothesise that assimilating multi-site daily $CH_4$ flux observations using the GRaB-AM framework can derive a set of optimised general parameters capable of representing various types of northern wetlands.



The present study's objective is to investigate the capacity of the GRaB-AM framework developed by Kallingal et al. (2023) for calibrating selected model process parameters in a more general multi-site setup. We aim to use the caliberated model parameters to assess the extent to which this optimisation improves the model's ability to estimate $CH_4$ emissions from various wetlands across northern latitudes and to estimate the total mean annual budget at larger scales. We also aim to estimate the posterior process and parameter uncertainties, as well as the posterior parameter correlations. The 14 different arctic, boreal and temperate wetland sites chosen for this study were selected to encompass diverse bioclimatic and geographical characteristics of wetlands. This deliberate selection aimed to endow the optimised parameters with the ability to represent a range of wetland types, independent of their distinct climatic and geographical features. We then perform additional validation simulations to evaluate the performance of the calibrated model against 5 different, independent validation sites to verify our above mentioned hypothesis. In the following, we first give a brief overview of the LPJ-GUESS model, the data used in the assimilation and the data assimilation methodology itself. The assimilation results are then presented and discussed in Sections 3 and 4 before we end with the conclusion (Sect. 5).

## 2 Data and methodology

### 2.1 LPJ-GUESS model

LPJ-GUESS represents the structure and dynamics of terrestrial ecosystems from local to global scales (Smith (2001), Smith et al. (2014)). The model combines basic eco-physiological features with detailed vegetation dynamics and canopy structure as used in forest gap models, and includes an interactive nitrogen cycle (Smith et al., 2014). In version 4.1, which we used for this study, global vegetation is grouped into 13 different co-occurring mixtures of Plant Functional types (PFTs) and five additional PFTs that can only exist on peatland stands. The model input data consists of climate parameters (mean daily air temperature, precipitation and incoming shortwave radiation), atmospheric $CO_2$ concentrations and soil properties. LPJ-GUESS simulates vegetation dynamics, ecosystem biogeochemistry, water cycling and energy and carbon fluxes on a daily time step. The peatland module in LPJ-GUESS contains detailed representations of wetland PFT characteristics and bio-geo-chemical processes such as estimation of peat temperature, hydrology and ecosystem exchanges, including $CH_4$ emissions.

### 2.1.1 Main process description in $CH_4$ module of LPJ-GUESS

A detailed description of the wetland and $CH_4$ emissions module is given in Wania et al. (2010) and in Kallingal et al. (2023). Here, we only briefly summarise the most important aspects of the module. The wetland peat in LPJ-GUESS is 1.5 m deep and is divided into an acrotelm with a thickness of 0.3 m with varying water table depth (wtd), and a permanently saturated catotelm. Peat hydrology and peat temperature in this layered structure depend on the composition of each layer and prevailing meteorological conditions. The five types of PFTs implemented in the wetlands are Sphagnum mosses, $C_3$ graminoids, evergreen and deciduous shrubs and a generic herbaceous cushion lichen moss. Shade mortality, inundation stress and daily desiccation stress are limiting factors for the existence and productivity of these PFTs. The basic concept of the $CH_4$ module



in LPJ-GUESS is a soil carbon pool distributed in proportion to the root distribution. This 'potential carbon pool' serves as the substrate for methanogens to produce $CH_4$. A portion of the soil carbon get transformed to $CH_4$ and/or $CO_2$ depending on the hydrological conditions. A fraction of the produced $CH_4$ is in dissolved form and the remainder is in gaseous form. A part of this $CH_4$ is oxidised by the oxygen in the soil and the other part is eventually transported to the atmosphere through either diffusion, plant-mediated transport or ebullition. As shown in Eq.1 the sum of the emissions through these three pathways constitutes the total $CH_4$ flux from the soil to the atmosphere (Wania et al. (2010), Kallingal et al. (2023)).

$$F_{CH_4} = CH_{4\,diff} + CH_{4\,plant} + CH_{4\,ebul} \tag{1}$$

where $F_{CH_4}$ is the total $CH_4$ flux, $CH_{4\,diff}$ is the $CH_4$ flux component from diffusion, $CH_{4\,plant}$ is the $CH_4$ flux component from plant-mediated transport and $CH_{4\,ebul}$ is the $CH_4$ flux component from ebullition.

### 2.1.2 Parameter selection

The parameters selected for optimisation in this study are shown in Table 1. For this study we have considered 10 out of 11 parameters calibrated by Kallingal et al. (2023) in their single-site optimisation. The parameter $w_{tiller}$, which is the plant tiller weight, is removed because it showed high correlation with $r_{tiller}$ in Kallingal et al. (2023).

**Table 1.** Parameters selected for the multi-site assimilation. Model prior values, prior standard deviation (std), units, and description of the parameters are given.

| Number | Parameter | Prior value | Prior std | Unit | Description |
|:---:|:---:|:---:|:---:|:---:|:---:|
| 1. | $R_{moist}$ | 0.4 | 0.396 | - | Moisture response in acrotelm |
| 2. | $CH_4/CO_2$ | 0.085 | 0.236 | - | $CH_4$ to $CO_2$ ratio |
| 3. | $f_{oxid}$ | 0.5 | 0.36 | - | Litter $CO_2$ fraction |
| 4. | $\phi_{tiller}$ | 70 | 36 | % | Porosity of tiller |
| 5. | $r_{tiller}$ | 0.0035 | 0.004 | $m$ | Radius of Tiller |
| 6. | $f_{air}$ | 0 | 4 | % | Fraction of air in peat |
| 7. | $por_{acro}$ | 0.98 | 0.06 | - | Acrotelm porosity |
| 8. | $por_{cato}$ | 0.92 | 0.076 | - | Catotelm porosity |
| 9. | $R_{moist-an*}$ | 0.025 | 0.04 | - | Moisture response in catotelm |
| 10. | $\lambda_{root}$ | 25.17 | 12 | $cm$ | Decay length of root biomass |

### 2.2 Flux sites and climate data

As mentioned in Sect. 1, for this study we selected 14 natural wetland sites for the assimilation and five additional wetland sites for validation (see Fig. 1, Table 2 and Table 4). The selection criteria for the sites were: 1) that they are located above $40°$ North, 2) that they include at least three years of consecutive $CH_4$ (at least the summer) measurements and meteorological





measurements available for the sites used for assimilation (same criteria, but, at least two years of measurements available for the sites used for validation) and 3) that they represent arctic, boreal or temperate ecosystems. We did not consider lakes, uplands, etc. as they are beyond the scope of the present study. The 19 sites are representative for a range of wetland types including fens, mires, bogs, marsh and a wet tundra (for validation).

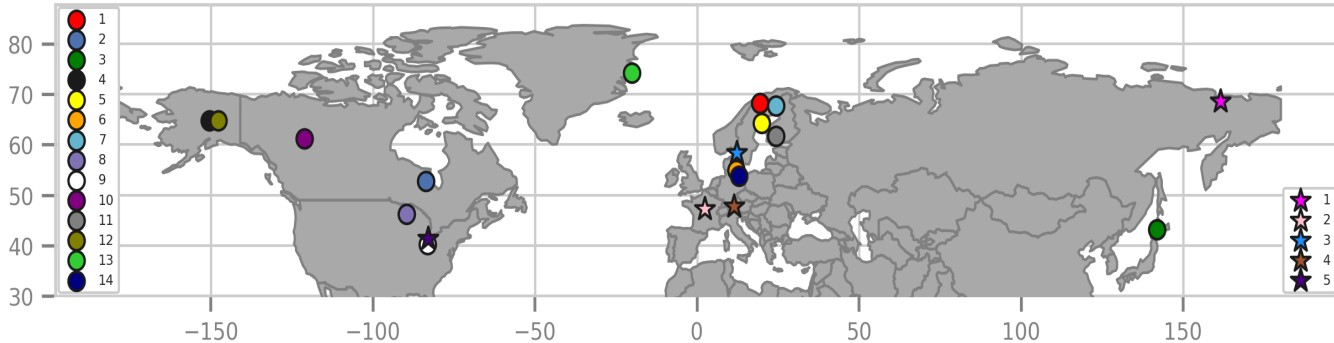

**Figure 1.** Location of measurement sites selected for the study. The 14 sites used for assimilation are indicated as circles and the 5 sites used for validation are indicated as stars. The numbers in the left and right side legends correspond to the numbers assigned to the sites in Table 2 and Table 4, respectively.

  In total we have used 18,437 data points of daily $CH_4$ flux observations for assimilation, spanning approximately 93
measurement-years. We did not consider any sites where the climate data had gaps of more than 14 days. In case of gaps smaller than 14 days the data are back-filled (by copying the values from the preceding cells) to ensure dataset continuity, as data with gaps cannot be used in the model as input. For validation, we utilised 5111 data points spanning a total of 14 measurement-years. For all sites, only the available $CH_4$ flux observations are compared with the model both for assimilation and validation, i.e., we do not use any gap-filled $CH_4$ fluxes. It should be noted that for the assimilation sites Att, Bon, Sco,
Uoa, and Zak, most of the data from winter months were missing, while for the validation sites Lgt and Wpt, most of the data from summer months were missing.





**Table 2.** Site information and data references of the 14 natural wetland sites used for assimilation. MAT refers to the mean annual temperature and MAPr to the mean annual precipitation. For Bib, Deg, Lom, Los, Ole, Sco, Sii, and Uao MAT and MAPr are extracted from the corresponding grid cells of the site locations of the WorldClim 2.0 gridded product (Fick and Hijmans, 2017), and for the rest of the sites, the information is taken from their references. The table also includes type, coordinates and climate zones of the wetlands, as well as the time period of data availability.

| No | Site | Abr. | Type | Location | Climate Zone | MAT (°C) | MAPr (mm) | Period | Reference |
|---|---|---|---|---|---|---|---|---|---|
| 1. | Abisko | Abi | Bog | 68.21°N, 19.03°E | Arctic | -0.7 | 304 | 2014-18 | Łakomiec et al. (2021) |
| 2. | Attawapiskat | Att | Fen | 52.70°N, -83.95°E | Boreal | -1.3 | 700 | 2011-20 | Todd and Humphreys (2018) |
| 3. | Bibai | Bib | Bog | 43.32°N, 141.81°E | Temperate | 6.7 | 1153 | 2015-19 | Ueyama et al. (2020) |
| 4. | Bonanza Creek | Bon | Bog | 64.69°N, -148.32°E | Boreal | -0.9 | 331 | 2014-17 | Euskirchen and Edgar (2020) |
| 5. | Degerö | Deg | Fen | 64.18°N, 19.55°E | Boreal | 1.2 | 523 | 2014-19 | Granberg et al. (2001) |
| 6. | Huetelmoor | Hue | Fen | 54.21°N, 12.17°E | Temperate | 10.2 | 572 | 2011-19 | Koebsch and Jurasinski (2020) |
| 7. | Lompolojänkkä | Lom | Fen | 68.00°N, 24.21°E | Boreal | -0.4 | 484 | 2006-15 | Lohila et al. (2020), Aurela et al. (2015) |
| 8. | Lost Creek | Los | Fen | 46.08°N, -89.97°E | Temperate | 4.8 | 833 | 2014-19 | Desai and Thom (2020) |
| 9. | Olentangy | Ole | Marsh | 40.02°N, -83.01°E | Temperate | 12.1 | 1120 | 2011-16 | Bohrer and Morin (2020) |
| 10. | Scotty Creek | Sco | Bog | 61.30°N, -121.29°E | Boreal | -2.8 | 414 | 2014-18 | Sonnentag and Helbig (2020) |
| 11. | Siikaneva | Sii | Fen | 61.83°N, 24.193°E | Boreal | 4.2 | 707 | 2005-15 | Rinne et al. (2018) |
| 12. | Uni. of Alaska | Uoa | Bog | 64.86°N, -147.85°E | Boreal | -2.9 | 611 | 2011-19 | Iwata et al. (2020) |
| 13. | Zackenberg | Zak | Fen | 74.30°N, -20.30°E | Arctic | -8.6 | 253 | 2006-20 | Scheller et al. (2021) |
| 14. | Zarnekow | Zar | Fen | 53.87°N, 12.88°E | Temperate | 9.7 | 426 | 2014-19 | Sachs et al. (2020) |

## 2.3 Data assimilation system

To find an optimal posterior parameter set we used an adaptive Rao–Blackwellised Markov Chain Monte Carlo Metropolis-Hastings (MCMC-MH) algorithm (Andrieu and Thoms, 2008) to iteratively reduce the model-data misfit in terms of a so-called cost function (see Eq. 2) that compares the modelled observable with the field measurements. As mentioned in Sect. 1 details of this search algorithm and its application to a single-site optimisation have been described in Kallingal et al. (2023). Efficient sampling from the target distribution requires a proposal distribution that correctly represents the dependence structure of the target, and to avoid manual tuning of the proposal they used an adaptive MCMC to tune the proposal distribution, where the Rao–Blackwellisation improves the adaptation step. The tuning improves the MCMC convergence speed and avoids cases of incomplete convergence (Andrieu and Thoms, 2008), especially for a complex non-linear model like LPJ-GUESS.

We assumed errors in observation and parameters in the form of Gaussian distributions yielding the cost function J (x),

$$J(x) = \frac{1}{2}\sum_{i=1}^{n}(Y_i - M_i(x))^t R_i^{-1}(Y_i - M_i(x)) + \frac{1}{2}(x - x_p)^t B^{-1}(x - x_p) \qquad (2)$$

where $Y_i$, $M_i(x)$, and $R_i$ are the CH$_4$ flux observations, model simulations, and the covariance matrix of the observation errors, respectively at the $i^{\text{th}}$ site, and $x_p$ are the expected prior parameters and $B$ is the prior parameter error covariance





matrix. Thus, the first term represents the model-data misfit weighted by the observation error covariances and the second term represents the prior information on the parameters weighted by the parameter error covariances.

Samples are generated by drawing $x_{\text{prop}}$ from a proposal distribution and then either accepting the proposed state ($x_i = x_{\text{prop}}$) or keeping the current state ($x_i = x_{i-1}$) based on the posterior probabilities. The probability of accepting the proposed state ($\alpha$) is generally computed as

$$\alpha = \min\left(1, \frac{P(x_{\text{prop}})}{P(x_{i-1})}\right) \hspace{6cm} (3)$$

Here, $P(x_{\text{prop}})$ is the posterior probability of the proposed state, and $P(x_{i-1})$ is the posterior probability of the current state, both computed using the cost function, Eq. 2. The acceptance probability ensures a balanced exploration of the parameter space, accepting states that improve the fit while allowing occasional exploration of less favorable regions (see Andrieu and Thoms (2008) for technical details and Kallingal et al. (2023) for the implementation).

**Table 3.** Data availability and threshold estimated for the base error values. The number of available flux observations from each site is also provided.

| No | Site | No. of obs. | Available data (%) | Threshold for base error ($gC\,m^{-2}d^{-1}$) | Error below the threshold ($gC\,m^{-2}d^{-1}$) |
|---|---|---|---|---|---|
| 1. | Abi | 1310 | 89.7 | 0.003 | 0.09 |
| 2. | Att | 1952 | 61.65 | 0.0012 | 0.036 |
| 3. | Bib | 815 | 60.1 | 0.01 | 0.3 |
| 4. | Bon | 560 | 60.0 | 0.01 | 0.3 |
| 5. | Deg | 1361 | 74.6 | 0.0057 | 0.15 |
| 6. | Hue | 2124 | 76.4 | 0.037 | 1.1 |
| 7. | Lom | 1682 | 51.3 | 0.011 | 0.32 |
| 8. | Los | 1472 | 83.0 | 0.0085 | 0.25 |
| 9. | Ole | 1135 | 74.6 | 0.0085 | 0.25 |
| 10. | Sco | 646 | 49.5 | 0.018 | 0.53 |
| 11. | Sii | 1547 | 44.1 | 0.01 | 0.3 |
| 12. | Uao | 1126 | 41 | 0.013 | 0.38 |
| 13. | Zak | 1294 | 26.67 | 0.007 | 0.21 |
| 14. | Zar | 1413 | 77.42 | 0.04 | 1.2 |

For optimisation we used the GRaB-AM with a chain length of 100,000 iterations, where each iteration involves one complete model run for all 14 sites. As mentioned above, daily averages of flux observations collected from the above-mentioned 14 sites are assimilated simultaneously. For each site, only the actual $CH_4$ flux observations, i.e., not gap-filled data are used to calculate the cost function. Having multiple sites in this framework, one crucial challenge was scaling the cost function to maintain a balanced representation of each site's contribution to the overall model-data misfit. This process is particularly

relevant when sites exhibit variations in the magnitude of their individual cost functions or when the number of observations





at each site differs significantly. Here, the scaling factors are carefully chosen to ensure an approximately equal representation of all sites in the cost function, regardless of their individual characteristics, and to ensure that each site has an equal influence on the optimisation outcome.

Considering the difficulty of calculating error correlations in the flux observations, we only considered errors in individual observations, i.e., we did not consider off-diagonal elements in specifying the observational error covariance matrix $R$ in the cost function (Eq. 2). Estimating the exact observation error for each site is again challenging. Assigning a constant percentage error for all measured values could introduce a bias, as it would result in high error values for measurements with high magnitudes and very low error values for observations with small magnitudes. To overcome this challenge, we followed the procedure introduced by Knorr and Kattge (2005) for the case of assimilating $CO_2$ eddy covariance observations and assign a threshold value set at 5% of the variance of the distributions of observations, calculated separately for each site. Values below this threshold are identified, and a uniform error is assigned to them (see Table 3) . An error of 30% is estimated for the observations greater than the threshold values. For the matrix $B$ in the Eq. 2, for each parameter a standard deviation of 40% of their possible range is assumed based on the expertise of LPJ-GUESS modellers.

## 2.4 Posterior estimation

After completing a full run of the MCMC chain, the posterior parameter error covariance matrix ($Bp$) is estimated from the prior error covariance matrices of the observations, R, and of the parameters, B, and the linearisation of the model at the minimum of the cost function, $J(M_\infty)$ , as described in Tarantola (1987),

$$Bp = [M_\infty^t R^{-1} M_\infty + B^{-1}]^{-1} \tag{4}$$

$Bp$ is then used to estimate the level of optimisation of each parameter and the sensitivity of the cost function to them. The posterior parameter uncertainties have been estimated from the square root of the diagonal elements of $Bp$. Large absolute values of posterior error correlations indicate that the observations do not provide independent information to distinguish the effects of a given parameter pair (Tarantola, 1987).

From the 100,000 samples yielded by the GRaB-AM framework 75% of this chain was discarded as burn-in. The remaining part of the chain, which we consider as converged to its stationary distribution was used for calculating the Maximum a Posteriori estimation (MAP) and posterior mean estimations. The posterior distributions of parameters are classified as 'well-constrained', 'poorly constrained', and 'edge-hitting' parameters. The well-constrained parameters are characterised by a clearly defined unimodal distribution with a low standard deviation. Conversely, poorly constrained parameters exhibit a relatively flat multimodal distribution with a large standard deviation. For a more precise estimation, we classified posterior parameter distributions as poorly constrained if the standard deviation exceeded 20% of the total range. Edge-hitting parameters cluster near one of the edges of their prior range, as described by Kallingal et al. (2023) and Braswell et al. (2005).





## 2.5 Experimental setup

The model is spun up for 500 years using Climate Research Unit (CRU) meteorological forcing data (University of East Anglia Climatic Research Unit , CRU), which includes daily measurements of air temperature, precipitation, and incoming shortwave radiation, to bring the model's state variables, i.e., the various carbon pools, to initial equilibrium. After spinning up, the model

was run for the 14 assimilation sites using local, daily meteorological forcing data collected directly at the sites. We have bias-corrected the CRU data for the gridcells in which the sites are located to enforce agreement with monthly mean values of the site-specific meteorological data. For this we have used at least two years of meteorological data that are recorded prior to the time period of the site flux observations that we used for the assimilation. Atmospheric $CO_2$ concentration, as described in McGuire et al. (2001) and updated until recent years using data from the NOAA Global Monitoring Laboratory (NOAA-GML)

is used as the $CO_2$ concentration input to the model.

Most of the $CH_4$ flux observations at the sites were available with a half-hourly resolution, but for the assimilation we used daily mean values corresponding to the LPJ-GUESS temporal resolution. Also, using daily values reduces the complexity of error correlations of half-hourly data and is better suited for a broad range time scale assimilation (Lasslop et al., 2008). Days with less than 50% of half-hourly $CH_4$ data availability were removed from the assimilation. To test whether the resulting

posterior parameters can enhance model performance for other individual wetlands within the study area, we designed a validation experiment that uses additional flux observations not included in the assimilation experiment. Independent flux observations from five different wetlands located in various parts of the study area were used for the experiment (see Sect. 2.2 and Table 4).

**Table 4.** Site information and data references of 5 natural wetland sites used for validation. MAT refers to the mean annual temperature and MAPr to the mean annual precipitation collected from their references. The table also includes the time period of data collected, the availability of data and the type and climate zones of the wetlands.

| No | Site | Abr. | Type | Location | Climate Zone | MAT (°C) | MAPr (mm) | Period | No. of Obs. | Available Data (%) | Reference |
|----|------|------|------|----------|--------------|----------|-----------|--------|-------------|--------------------|-----------|
| 1. | Chersky | Che | Wet Tundra | 68.61°N, 161.35°E | Arctic | -9.8 | 200 | 2014-2017 | 923 | 84 | Merbold et al. (2020) |
| 2. | La Guette | Lgt | Fen | 47.32°N, 2.28°E | Temperate | 11.07 | 650 | 2017-2019 | 227 | 31 | Jacotot et al. (2020) |
| 3. | Mycklemossen | Myk | Bog | 58.36°N, 12.16°E | Hemi-boreal | 6.9 | 802 | 2016-2019 | 1095 | 100 | White et al. (2023) |
| 4. | Schechenfilz N. | Sfn | Bog | 47.80°N, 11.32°E | Temperate | 8.28 | 700 | 2012-2015 | 700 | 64 | Schmid and Klatt (2020) |
| 5. | Winous Point N. | Wpt | Marsh | 41.48°N, -82.99°E | Temperate | 11.4 | 900 | 2011-2014 | 477 | 44 | Chen and Chu (2020) |

Additionally, we conducted a mean annual budget estimation for all wetlands above 45°N and for all wetlands above 60°N

using the optimised parameters. The model was run using the peat fraction map PEATMAP (Xu et al., 2018) and daily gridded climate data from the Climatic Research Unit-Japanese Reanalysis (CRU-JRA) (Harris et al., 2020) as model inputs. CRU-JRA data can be accessed at https://rda.ucar.edu/datasets/ds628.0/. The results were then compared with the output of the JSBACH-HIMMELI model, as described in Zhang et al. (2023) (hereafter referred to as JSBACH-H), for regions above 45 °N (for easier comparison with other literature) as well as with the estimates from the Global $CH_4$ Budget of the Global Carbon Project

(GCP) (Saunois et al., 2024) for above 60°N.





## 2.6 Statistical metrics

The following statistical metrics are used, among others, to analyse the results. The total reduced chi-square ($\chi^2$) is used to assess the goodness of the fit between the flux data and the model outputs. We calculated it by dividing twice the value of the cost function by the number of observations as,

$$\chi^2 = \frac{2J(x)}{N} \tag{5}$$

where $J$ is the cost function, $x$ corresponds to the parameters, and $N$ is the number of observations.

The Root Mean Square Error (RMSE) is estimated to quantify the average error between predicted and observed flux data as,

$$\text{RMSE} = \sqrt{\frac{1}{N}\sum_{i=1}^{N}(Y_i - M_i)^2} \tag{6}$$

where $N$ is the number of observations, $Y_i$ is the observed value, and $M_i$ is the predicted value.

An uncertainty estimation was conducted assuming independence between parameter uncertainty and model uncertainty, using the following equation:

$$\sigma_{total} = \sqrt{\sigma_{\text{model}}^2 + \sigma_{\text{param}}^2} \tag{7}$$

where $\sigma_{\text{model}}$ is the model structural uncertainty estimated from the standard deviation of the prior and posterior residuals (Desroziers et al., 2005). $\sigma_{\text{param}}$ represents the contribution of parameter uncertainty to the overall uncertainty in observation space, estimated from the 95 % credible interval of the parameters and the standard deviation of total sums of the model prediction by taking into account both the parameter uncertainty from the MCMC sampling and the variability in the model predictions. The calculation is performed as follows:

$$\sigma_{\text{param}} = \frac{\sigma_{\text{predic}}(CI_{\text{upp}} - CI_{\text{low}})}{1.96} \tag{8}$$

where, $\sigma_{\text{predic}}$ is the the standard deviation of total sums of the model prediction over MCMC runs, $CI_{\text{upp}}$ and $CI_{\text{low}}$ are the upper and lower bounds of the credible intervals of the parameters, and the factor 1.96 is the conversion factor to convert the 95 % credible interval to a standard deviation assuming a Gaussian distribution.

In addition, we used skewness and kurtosis estimates of the posterior parameter Probability Density Functions (PDFs) to
describe their structure. We also calculated summer and winter anomalies by subtracting the mean values for April through September (summer) and October through March (winter) from the corresponding time series to estimate the variability in the seasonal cycle of the observed and modeled $CH_4$ fluxes.

 

## 3 Results

### 3.1 Posterior parameter distributions

The PDFs of the parameters from the MCMC chains after the burn-in are displayed in Fig. 2. All parameters, except for $\phi_{\text{tiller}}$, $f_{\text{air}}$ and $por_{\text{acro}}$, are well-constrained with only one peak in the PDF. However, the parameters $\phi_{\text{tiller}}$, $f_{\text{air}}$ and $por_{\text{acro}}$ are rather poorly constrained, exhibiting some clustering around multiple peaks in the PDF and having large standard deviations. The $R_{moist}$, $f_{air}$, and $\lambda_{root}$ showed high skewness and kurtosis. However, the smaller kurtosis values of $f_{oxid}$, $por_{cato}$, and $R_{moist_{an}}$, along with their low skewness, indicate that they closely resemble Gaussian distributions. The remaining four

parameters showed low skewness, suggesting agreement with a Gaussian distribution, but with very low kurtosis indicating somewhat flatter distributions than a Gaussian distribution. The figure shows that, except for $\lambda_{\text{root}}$, none of the parameters exhibited edge-hitting behavior, indicating that the hypothetical boundaries assigned for each parameter align well with the model structure. The parameter $\lambda_{\text{root}}$ finds its solution nearly at the lower edge. Most parameters displayed posterior solutions far from their prior values, i.e., the prior values where outside the posterior mean estimate $\pm 1\sigma$, except for $\phi_{\text{tiller}}$. For $\phi_{\text{tiller}}$,

both the MAP and posterior mean appeared close to the prior value, i.e., within $1\sigma$ of the posterior mean estimate.

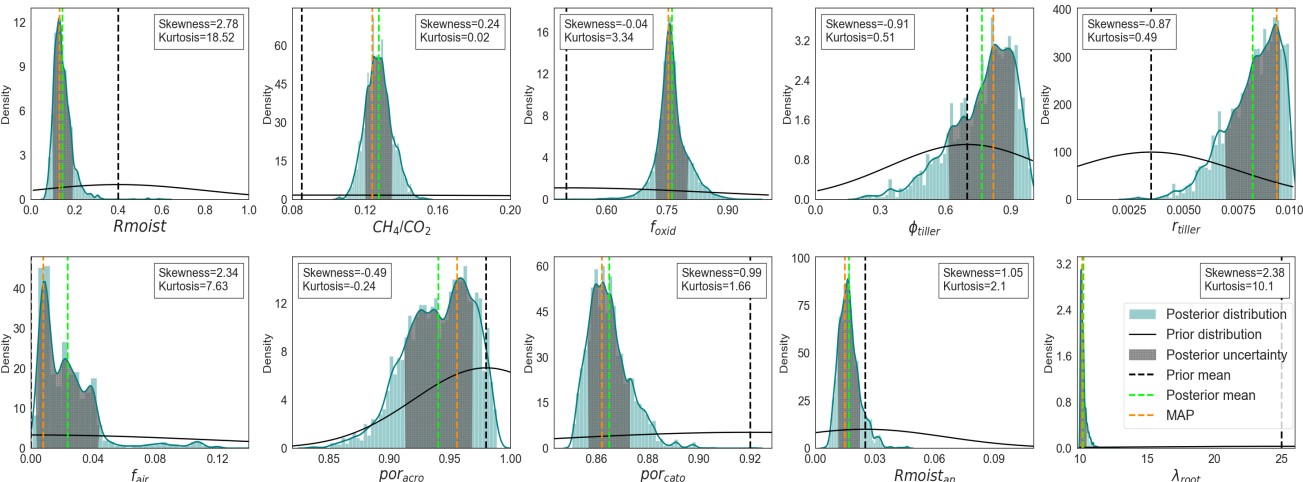

**Figure 2.** Probability distribution functions (PDFs) of the posterior obtained after the GRaB-AM experiment. The blue curves are smoothed Gaussian kernel estimates on the posterior histograms, while the black curves represent the prior distributions. The dotted vertical lines in green, orange, and black correspond to the posterior mean, MAP, and prior means, respectively. The shaded grey area in the distributions represents the $1\sigma$ error estimate of the PDFs. Skewness and kurtosis values for each posterior distribution are provided in rectangles.

### 3.2 Posterior parameter estimates and cross-correlation

The posterior parameter values estimated from the MAP and the posterior mean estimate, along with their standard deviations, are presented in Table 5. The MAP and posterior mean estimates of all parameters remained close to each other, except for




$\phi_{tiller}$, $r_{tiller}$, and $f_{air}$. The parameters $\lambda_{root}$ and $\phi_{tiller}$ exhibited high standard deviations in their posterior distributions,
indicating greater uncertainty. In contrast, the parameters CH$_4$/CO$_2$, tiller, por$_{cato}$, and $r_{most}$ showed low standard deviations,
suggesting more precise posterior estimates.

**Table 5.** Posterior parameter value estimate of the GRaB-AM. The prior values, maximum a posteriori (MAP), posterior mean, posterior standard deviations (std) are given. The cost function values of prior and posterior estimates are also given.

| | Parameter | | | | | | | | | | |
| --- | --- | --- | --- | --- | --- | --- | --- | --- | --- | --- | --- |
| | $R_{\text{moist}}$ | $CH_4/CO_2$ | $f_{\text{oxid}}$ | $\phi_{\text{tiller}}$ | $r_{\text{tiller}}$ | $f_{\text{air}}$ | por$_{\text{acro}}$ | por$_{\text{cato}}$ | $R_{\text{moist\_an}}$ | $\lambda_{\text{root}}$ | **Cost value (w)** |
| Start prior vals. | 0.3 | 0.2 | 0.6 | 0.8 | 0.003 | 0.001 | 0.9 | 0.87 | 0.04 | 35 | 4897.95 |
| MAP | 0.13 | 0.12 | 0.75 | 0.82 | 0.0092 | 0.008 | 0.95 | 0.86 | 0.017 | 10.25 | 221.01 |
| Posterior mean | 0.15 | 0.14 | 0.76 | 0.77 | 0.0081 | 0.023 | 0.94 | 0.86 | 0.017 | 10.25 | 227.30 |
| Posterior std $\mp$ | 0.045 | 0.007 | 0.027 | 0.15 | 0.0012 | 0.02 | 0.027 | 0.008 | 0.005 | 0.23 | |

A cross-correlation plot (see Fig. 3) shows the correlation between all ten parameter pairs after optimisation to examine potential optimisation issues due to parameter correlation. High positive or negative correlations suggest that these parameters may convey similar information, and one of them might be redundant in further studies. The results show that not many 250 parameters have strong positive or negative correlations, except for the correlation between $CH_4/CO_2$ and $Rmoist_{\text{an}}$, which has a high negative correlation of -0.82. Many slight positive correlations are observed, with a few pairs like por$_{\text{acro}}$ and por$_{\text{cato}}$, por$_{\text{cato}}$ and $f_{\text{air}}$ having comparatively higher values. A detailed discussion of the parameter correlation and their possible impacts on the posterior estimates is given in Sect. 4.1.







**Figure 3.** Posterior correlations between parameters derived from the GRaB-AM optimisation. In the upper triangle of the figure, negative correlations are depicted in blue and positive correlations are shown in red. The numerical labels on the upper triangle correspond to values of Pearson's correlation coefficient. The diagonal panels exhibit 1-D histograms for each model parameter. The lower triangle displays two-dimensional marginal distributions for each parameter. The grey dots on the marginal distributions represent the parameter values obtained from the posterior GRaB-AM chain. The ranges of the distributions are labeled on the left and bottom of the figure.

## 3.3 Performance of the optimisation

The site-wise data-model misfit, presented in terms of RMSE, both before and after optimisation, along with the average RMSE for all sites combined is presented in Fig. 4. Most sites demonstrate a substantial reduction in RMSE, with many achieving over



50 % improvement. The unweighted cost function values for each of the 14 sites individually and their collective sum, along with the corresponding RMSE and reduced $\chi^2$ values, are presented in Table B1 in the appendix. The estimated prior cost function value was 1,763,294.9. After optimisation the value changed to 79,296.4 (around 5 % of the prior) for the posterior mean estimate. The total $\chi^2$ value is 8.6, accounting for measurement and parameter uncertainty. Notably, none of the sites exhibit overconfidence, with individual $\chi^2$ values larger than 1, except for the site Hue, which shows a $\chi^2$ value slightly below, but close to 1. The median RMSE reduction is 52.2%, and the $\chi^2$ value is 6.8. Taking these values as thresholds, the sites Abi, Att, Sii, and Lom show significant reductions in RMSE, but with high $\chi^2$ values. In contrast, sites such as Bib, Deg, Hue, Sco, Zak, and Zar exhibit low RMSE reductions but low $\chi^2$ values. The sites Bon, Los, Ole and Uoa display both high RMSE reductions and low $\chi^2$ values. None of the sites are characterised by low RMSE reduction and a high $\chi^2$ value.

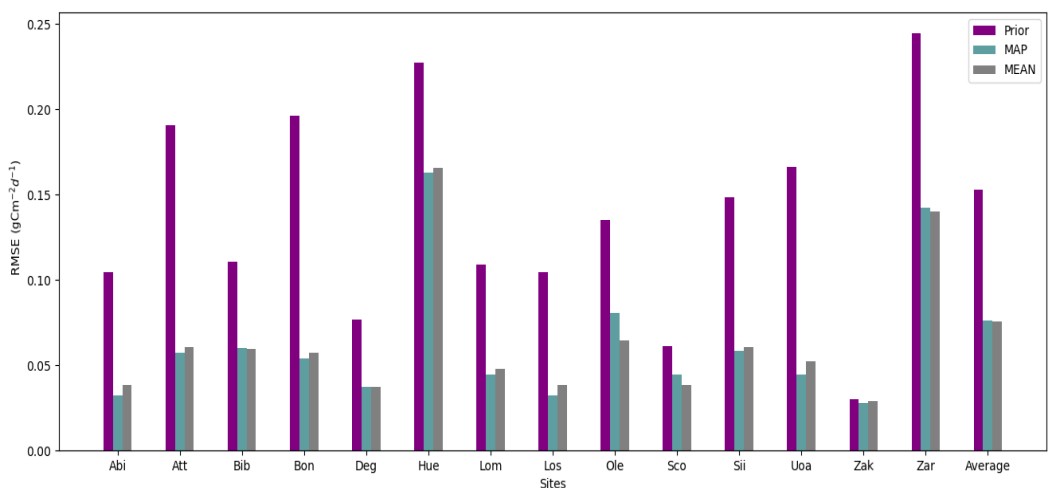

**Figure 4.** Prior and posterior Root Mean Square Error (RMSE) estimates are provided for each of the 14 sites individually, along with the combined average values. In the figure, purple, cyan, and grey bars represent the RMSE corresponding to the prior estimate, Maximum A Posteriori (MAP) estimate, and posterior mean estimate, respectively.

## 3.4 Impact of the optimisation

The observed $CH_4$ flux data available from all 14 sites collectively provide a total of 888.8 $gCm^{-2}$. The prior sum estimate for the corresponding data was 1957 $gCm^{-2}$. This value reduced to 850.9 $gCm^{-2}$ after optimisation. A reduction of approximately 56.52 % from the prior to the posterior $CH_4$ flux was observed. Compared to the observation, the posterior resulted only a slight underestimation of -38.1 $gCm^{-2}$. Time-series of the annual sums of $CH_4$ emissions at four of the 14 sites used in this study are shown in Fig. 6a. The time-series of the remaining sites are shown in appendix B2, Fig. B2 for convenience. All four sites shown in Fig. 6a exhibited a better fit to the annual budget after the optimisation. Particularly noteworthy is the site Bib, which had a prior estimation in 2016 significantly deviating from the observed values. This discrepancy was corrected in the posterior estimate, and the annual posterior $CH_4$ emissions at all four sites align well with the flux observations after the optimisation. Figure B2 shows that the sites Att, Los, Uoa, and Zar have shown improvement in all the years assimilated.





Deg improved in more than half of the years assimilated. Hue improved in 2013 but did not improve in the other years. Bon improved in two out of three years but was slightly overestimated in 2017. For Zak, although the prior and posterior merged in some years, it showed improvement in almost all years. Lom showed improvement in 2010, 2012, 2013, and 2014. Sco showed improvement in 2016 and 2017, which account for half of the total assimilated years. Figure 6b displays the mean annual sums

of $CH_4$ estimated at all 14 sites. The figure illustrates that the highest contribution came from the sites Hue and Zar, which have the highest mean annual temperature (MAT) as compared to other sites. The lowest contributions are from Zak and Abi, which have below-zero MAT. Abi, Att, Los, Ole, Sco, Sii, Uoa, Zak, and Zar showed improvement in the mean annual budgets after the optimisation. The remaining sites did not show an improvement.

Table 6 presents the total uncertainty for each site and the total uncertainty estimated for all sites together (see Sect. 2.6

for technical details). The total uncertainty of the posterior $CH_4$ flux estimates for all the sites together was 0.19 $gCm^{-2}d^{-1}$, whereas for the prior fluxes it was 0.36 $gCm^{-2}d^{-1}$. This results in a reduction of the total uncertainty of around 50 % after the optimisation. Comparing the prior and posterior RMSE (Fig. 4) and the uncertainty reduction, it can be concluded that the more constrained sites, such as Abi, Att, Bon, and Uoa, exhibited high uncertainty reduction. Notably, Abi and Att, which had the highest prior RMSE, showed a reduction of uncertainty of around 95 % and 82 %, respectively. A low reduction in

uncertainty was mainly observed in sites that demonstrated a low reduction in RMSE. In contradiction to this, even though the RMSE reduction observed in the case of Zak is very small, this site showed an uncertainty reduction of around 33 %.

**Table 6.** Prior and posterior total uncertainty estimates ($\sigma$ ($gCm^{-2}d^{-1}$)). Total parameter and model uncertainty estimates separately for each site and collectively for all sites combined are shown.

| Site | Prior Unc. | Posterior Unc. | Site | Prior Unc. | Posterior Unc. |
|------|-----------|----------------|------|-----------|----------------|
| Abi | 0.18 | 0.01 | Los | 0.04 | 0.03 |
| Att | 0.11 | 0.02 | Ole | 0.02 | 0.019 |
| Bib | 0.08 | 0.04 | Sco | 0.10 | 0.02 |
| Bon | 0.15 | 0.02 | Sii | 0.05 | 0.04 |
| Deg | 0.03 | 0.02 | Uaf | 0.08 | 0.03 |
| Hue | 0.13 | 0.12 | Zak | 0.03 | 0.02 |
| Lom | 0.03 | 0.027 | Zar | 0.11 | 0.10 |
| | | | Total Unc. | 0.36 | 0.19 |

Figure 5 illustrates the time-series of observed, prior, and posterior fluxes for all 14 sites considered in the assimilation. The posterior model continued to overestimate emissions at the Abi, Att, Los, Ole, Sii, and Uoa sites. For Bib (except for 2015), Deg, Hue, and Zak emissions were consistently underestimated. The remaining four sites demonstrated reasonably

good agreement of the posterior estimates with the flux observations. The model adeptly captures the seasonal cycles of $CH_4$ emissions from all wetlands, both for flux estimates using the prior and the posterior parameter values. Generally, no significant phase shift is observed in either the prior or the posterior estimates. However, for Bib, both prior and posterior estimates exhibit a slight phase shift to early summer, while the sites Hue, Los and Zar show a similar shift to late summer.



**Figure 5.** The CH$_4$ simulation from the LPJ-GUESS model from 14 different wetland sites (green dots) after optimising with the GRaB-AM algorithm. The black dots are the real CH$_4$ flux observations from corresponding wetlands. The red dots are the prior simulation with the prior model parameters used to start the MCMC chain. Three days of running averages are calculated from the original time series, and from most of the figure a few outliers on the vertical axis have been removed for better visualisation.



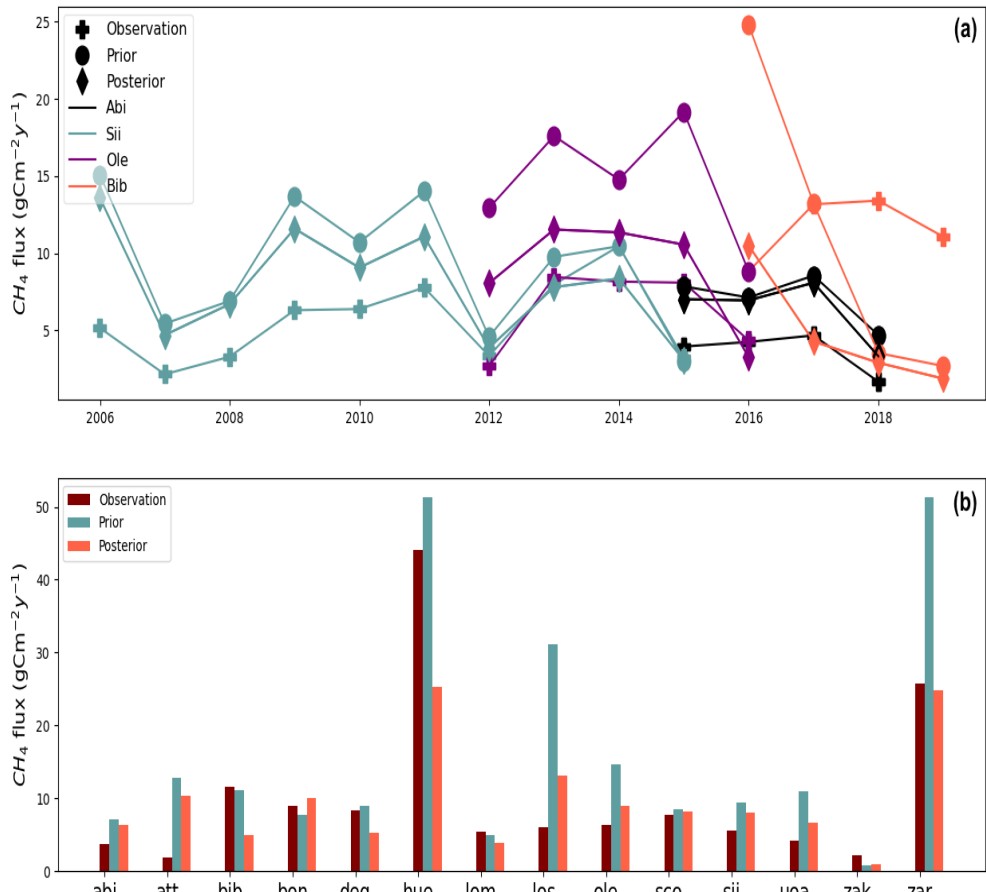

**Figure 6.** Figure (a) displays the annual sum estimation of CH$_4$ from four out of 14 sites used in the study. The sites are represented in different colours with distinct markings to distinguish between Observation, Prior, and Posterior. Figure (b) presents the mean annual sums of CH$_4$ estimated for all 14 sites used in this study as bar charts. The averages of the flux observations are calculated with only the available daily averages used for the assimilation.

### 3.4.1 Changes in component contribution

Changes in the component-wise estimation of ebullition, diffusion, and plant-mediated transport before and after optimisation is illustrated in Fig. 7. The inner circles represent the priors, while the outer circles represent the posterior model estimates. The optimised parameters are constrained differently for different components across sites. There was no general trend of any one of the three transport mechanisms being dominant after optimisation. Regarding the prior estimate, all sites but Hue and Att had significant contributions from all three emission components. Hue and Att had a very minor contribution from plant-305 mediated transport. After optimisation, zero contributions from plant-mediated transport were estimated for both these sites. Interestingly, for the site Los, the majority of the prior was contributed by plant-mediated transport and ebullition. However, in the posterior, ebullition contributed very little and was taken over by diffusion. Furthermore, many sites showed the dominance





of only two components after optimisation. It was consistently observed that the third component was suppressed, regardless of the nature or climatic conditions of the site.

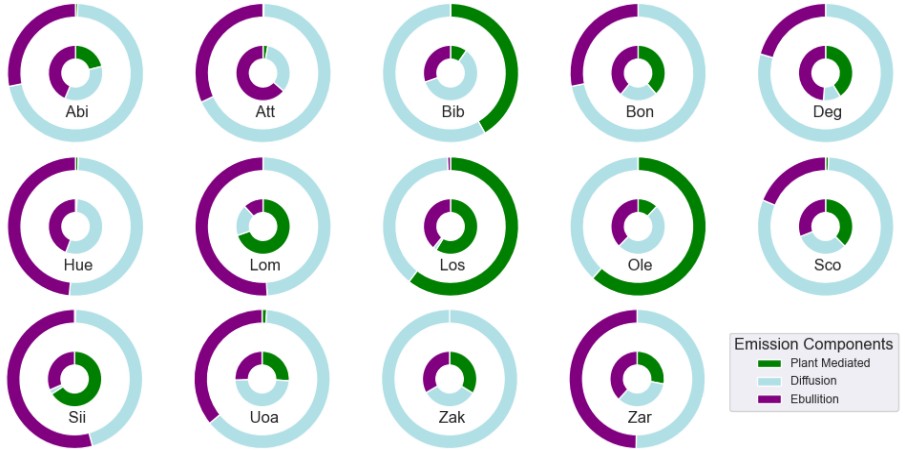

**Figure 7.** Component-wise percentage contribution of $CH_4$ to the total modelled emission for all 14 sites is presented separately. The inner circle represents the prior estimate, and the outer circle represents the posterior estimate.

Another interesting observation pertains to the site Zak, an arctic Fen with a very low mean annual temperature (MAT), where nearly equal contributions from all three components were observed in the prior. The posterior, however, showed that nearly all emissions were from diffusion, with very little contributions from the two other components. The RMSE estimate for this site indicates a very low reduction compared to other sites, suggesting that the optimisation did not perform well in constraining this site.

**3.4.2   Summer and winter anomalies**

Figure 8 illustrates the mean annual summer and winter emissions for all sites and their corresponding standard deviations. Figure B1 in the appendix shows their 'summer' and 'winter' anomalies. It should be noted that the winter mean and anomaly estimation for the sites Att, Bon, Sco, Uoa, and Zak was conducted with only a very limited number of available data points, as most of them were missing. Conversely, it should be taken into account that proper winter measurements were not carried out at
these sites, given the almost negligible emission estimates during the winter months due to their extremely cold temperatures. For all these sites, the mean annual temperature (MAT) was estimated to be below zero (refer to Table 2 and the corresponding site references).





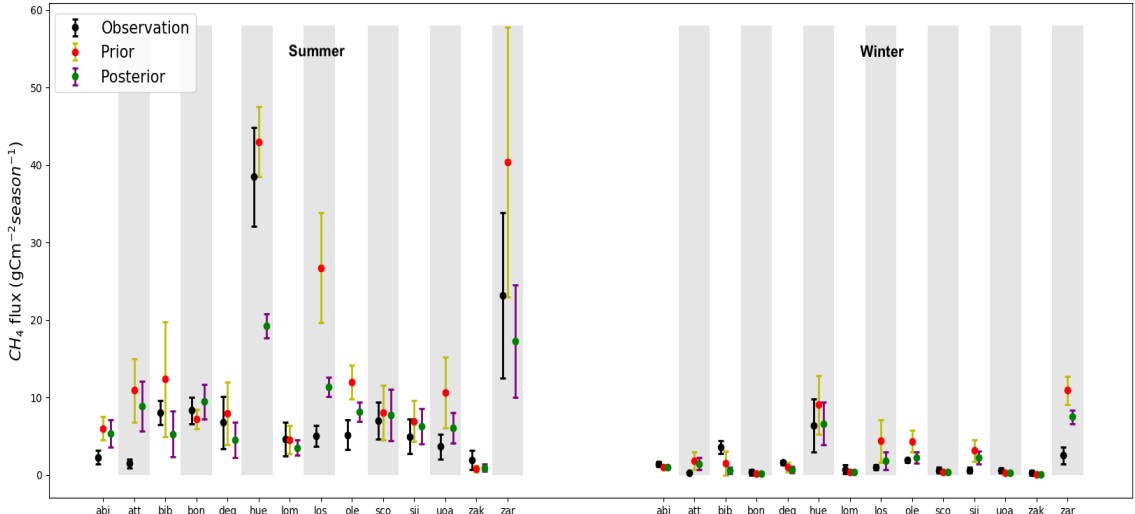

**Figure 8.** Mean annual summer (left side) and winter (right side) emissions for all sites, along with their corresponding standard deviations. The dots represents mean seasonal values, and error lines indicates their 1 standard deviation.

After the optimisation, both mean annual summer and winter emission estimates for sites Abi, Att, Los, Sco, Sii, and Zar exhibited improved agreement with the flux observations. The sites Uoa and Bib showed improvement in summer but not in winter whereas the sites Bon, Hue, Lom, and Ole improved in winter but not in summer. The sites Zak and Deg did not show any significant improvement in either season. For the site Deg, the prior estimates were closer to the flux observations than the posterior, and for the site Zak no significant changes were observed. Summer emissions in Zak were underestimated by both the prior and posterior models. This may be attributed to the relatively low (below zero) mean annual temperature (MAT) of -8.6 at this location, variation of summer months with latitude, and the seasonality of decomposition. In contrast, for the sites Abi and Att with negative MAT, the model tends to overestimate summer emissions, but the observed average summer emissions were comparatively lower. However, despite a MAT of -2.9 °C at the site Sco and -2.6 °C at the site Uoa, both these sites demonstrated relatively high summer emissions, which were better captured by the model using the posterior parameter values. Sites with a higher MAT, such as Hue, Bib, and Zar, exhibited the highest summer emission values. Although, the site Ole, which has the highest MAT of 12.1 °C, displayed comparatively lower summer emissions. This difference could be influenced by the substantial MAPr of 1120 mm at the Ole site.

For most of the sites, the observed winter annual mean was very close to zero, except for Bib, Deg, Hue, Ole, and Zar. For Hue, Ole, and Zar, the posterior estimate showed better performance in capturing the seasonal trends in flux observations than the prior. The site Hue exhibited high winter emissions. When the air temperature input for this site was estimated, it showed a mean value of 3.8 °C in winter months. Overall, although the majority of sites showed improved estimation of winter and summer emissions, some of the sites remained unchanged or did not perform better than before. The estimation of the standard deviation for summer and winter months showed a reduction for all sites after optimisation.



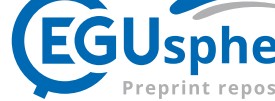

## 3.5 Validation of optimised parameters and large scale simulation

Table 7 and Figure 9 show the results of the validation experiment explained in Sect. 2.5. A collective reduction of 58.8% in RMSE was observed across all the validation sites. The RMSE decreased from 0.4 in the prior simulation to 0.16 in the posterior simulation. The total sum of flux observations for all five sites was 319.4 $gCm^{-2}$. Corresponding posterior simulation resulted in a total sum of 421.3 $gCm^{-2}$, compared to 521.9 $gCm^{-2}$ in the prior simulation. This represents a 19.3% decrease in the posterior estimate compared to the prior. Similarly, the prior $\sigma$ was 0.61, which was reduced to 0.17, representing a 72.13% reduction.

**Table 7.** RMSE reduction and the changes in the total emission estimate (in %) for the validation sites, along with their prior and posterior uncertainty ($\sigma$) estimates.

| Site | RMSE Reduction (%) | Change in Total Estimation (%) | Total Prior $\sigma$ | Total Posterior $\sigma$ $gCm^{-2}d^{-1}$ |
|------|------|------|------|------|
| Che | 31.2 | -21.5 | 0.29 | 0.19 |
| Lgt | 0.6 | 4.9 | 0.10 | 0.10 |
| Myk | 59.8 | 43.6 | 0.17 | 0.09 |
| Sfn | 81.1 | 38.5 | 0.77 | 0.16 |
| Wpt | -0.23 | 31.8 | 0.2 | 0.18 |
| Total | 58.8 | 19.3 | 0.61 | 0.17 |

The posterior estimates for four of the five sites showed improved RMSE. For the sites Sfn, Myk, and Che, reductions of 81.1%, 59.8%, and 31.2% were observed, respectively. The RMSE improvement for the site Lgt was negligible, with a value of 0.6%. The site Wpt, a temperate marsh (note that marshes were not included in the the sites used for assimilation), exhibited a very slight increase in RMSE. The total prior model-data mismatch of $CH_4$ estimated at this site during the time period was 72.5 $gCm^{-2}$, which increased to 78.98 $gCm^{-2}$ after optimisation. Including Wpt, the posterior estimates of all these sites appeared improved in terms of $\sigma$ reduction. Wet tundras were not used for assimilation; however, the site Che, a wet tundra used for validation, demonstrated a remarkable 31.2% reduction in RMSE, along with a significant reduction in total posterior $\sigma$, from 0.29 to 0.19. For all the sites except Che, the prior was overestimating the posterior flux. For Che, there was an increase of 21.5% in the posterior estimate.



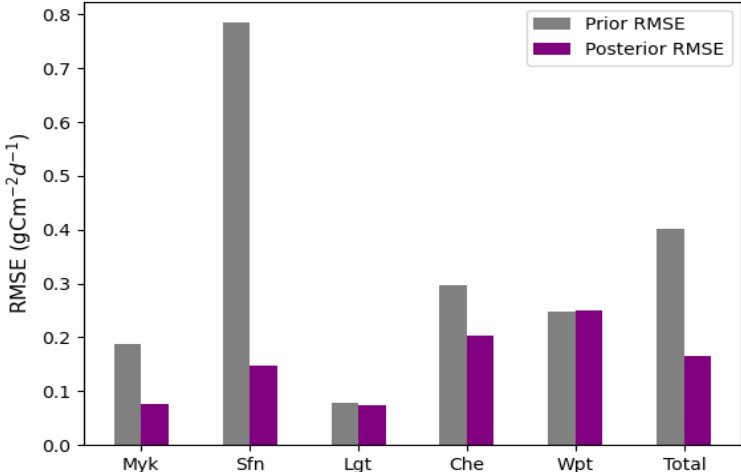

**Figure 9.** Prior and posterior Root Mean Square Error (RMSE) estimates are provided for each of the 5 validation sites individually, along with the combined average values. Purple, and grey bars represent RMSE corresponding to the prior estimate and posterior mean estimate, respectively.

Figures 10 and 11 show the results of large-scale simulations of $CH_4$ emissions from wetlands located above 45°N. The Fig. 10 includes prior and posterior simulations and their differences, as well as the fluxes from JSBACH-H simulations for

the same domain. The maps in the top row indicate that both the prior and posterior simulations identify hotspots of emission in North America, Canada, Russia, and Europe, though the total posterior estimate is clearly smaller compared to the prior. However, examining the differences of prior and posterior emission illustrated in the bottom left map reveals that the smaller posterior emissions are not systematic over the whole simulation domain. Essentially the magnitude of the emissions at the hotspot locations is reduced in the posterior whereas surrounding areas of low emissions in the prior are slightly enhanced

in the posterior. The bottom right map displays the $CH_4$ simulation from JSBACH-H, with a resolution 1.875°X 1.875°, for comparison.

It is evident that both LPJ-GUESS and JSBACH-H exhibit similar patterns in high and low emissions. The total mean emission over 2010-2020 was 32.32 Tg C y$^{-1}$ for the LPJ-GUESS prior simulation and 28.16 Tg C y$^{-1}$ for the posterior simulation. For JSBACH-H, the mean emission over 2010-2016 was 15.66 Tg C y$^{-1}$. The posterior simulation of LPJ-GUESS

is closer to the result from JSBACH-H but still much larger. When comparing $CH_4$ emissions from wetlands above 60 °N, the agreement between the posterior LPJ-GUESS estimate (7.46 Tg C y$^{-1}$) and the JSBACH-H estimate (7.62 Tg C y$^{-1}$, mean over 2010–2016) is much closer. However, the LPJ-GUESS prior estimate (8.69 Tg C y$^{-1}$) aligns better with the Global $CH_4$ Budget estimate from the GCP (9.0 Tg C y$^{-1}$, mean over 2010–2019).





**Figure 10.** Total mean annual $CH_4$ estimates above 45 °N. Top Left: Prior $CH_4$ Emissions from LPJ-GUESS averaged from 2010 to 2020 with a spatial resolution of 0.5°x 0.5°. Top Right: Posterior $CH_4$ Emissions from LPJ-GUESS averaged from 2010 to 2020 with a spatial resolution of 0.5°x 0.5°. Bottom Left: Difference between the prior and posterior emissions from LPJ-GUESS. Bottom Right: $CH_4$ Emissions from JSBACH-H averaged from 2010 to 2016 with a spatial resolution of 1.875°x 1.875°.





## 4   Discussion

In this study we have conducted detailed examination of the posterior results to assess parameter distributions, uncertainties, error correlations, changes in flux components, model-observation fit, as well as the changes in summer and winter anomalies. The well-constrained seven parameters discussed in Sect. 3.1 with single-peaked PDFs indicate strong convergence and effective estimation, suggesting the model is sensitive to the data for these parameters. However, the poorly constrained parameters with multiple peaks highlight areas where the model might need refinement or where additional types of observations could

improve parameter estimation. The ranges of posterior parameter distributions indicate a successful search within the permitted parameter ranges (Fig. 2). The non-Gaussian behavior of the parameters $\phi_{\text{tiller}}$, $f_{\text{air}}$, and $por_{\text{acro}}$ described in Sect. 3.1 can still be considered acceptable, as the posterior distribution is expected to be perfectly Gaussian only under the assumption of a linear model. Beyond the inherent nonlinearity of LPJ-GUESS, this behavior also suggest that the assimilated fluxes provide more complex information about the wetland processes than initially assumed or that there are complex interactions between the

model parameters. This complexity may warrant further investigation into the model structure. It also indicates that the model could benefit from accounting for errors that place less weight on extreme observations, allowing for a smoother response curve from the model.

### 4.1   Parameter correlations and their implications

$R_{moist}$ and $Rmoist_{\text{an}}$: These parameters are related to the moisture response in the acrotelm and catotelm, respectively. A

smaller posterior value of $R_{moist}$ and $R_{moist_{an}}$ would result in a slower soil carbon turnover time in the posterior model, leading to slightly less carbon available for CH$_4$ production. Although this could decrease the total decomposed carbon in the soil, the strong negative correlation between $R_{moist_{an}}$ and $CH_4/CO_2$, and the weak negative correlation between $R_{moist}$ and $CH_4/CO_2$, indicate that the decrease in these parameters has influenced the increase in the CH$_4$ fraction from this reduced amount of decomposed soil fraction. This indicates that other factors such as water table depth, availability of oxygen, soil

temperature, etc., might have influenced the CH$_4$ production.

$CH_4/CO_2$: This parameter represents the ratio of CH$_4$ to CO$_2$.The parameter was slightly increased to a value of 0.14 from 0.085 following optimisation. This indicates a comparatively higher CH$_4$ emission fraction from the total decomposed carbon. All the parameters except for $R_{moist}$ and $R_{moist_{an}}$ discussed above showed weak positive or neutral correlations with $CH_4/CO_2$. The weak positive correlations observed between most parameters and the $CH_4/CO_2$ ratio suggest that, while

these factors may not be the primary drivers of CH$_4$ production, they still contribute to the increases in the ratio.

$f_{\text{oxid}}$: The fraction of oxidised CH$_4$, utilising available oxygen in the soil, is represented by the parameter $f_{\text{oxid}}$. The posterior parameter value (0.76) is increased compared to the prior (0.5). This indicates that a substantial fraction of the produced CH$_4$ will get oxidised, while the remaining CH$_4$ (24%) will get transported to the atmosphere. This parameter has shown very low correlation with the other parameters and zero correlation with $r_{\text{tiller}}$ and $por_{\text{cato}}$.

$\phi_{\text{tiller}}$ and $r_{\text{tiller}}$: The posterior parameter values estimated for $\phi_{\text{tiller}}$ and $r_{\text{tiller}}$ are higher than the prior values, 0.77 and 0.0081, respectively. With aerenchyma tissues having more porous space and a larger radius, the plant-mediated transport of CH$_4$ to the



atmosphere is facilitated. However, through the same spacious aerenchyma tissues, plants also have the potential to transport more $O_2$ to the soil. This potential increase in the transport of $O_2$ to the soil could be a reason for increase in $f_{\text{oxid}}$, considering the slight positive correlation observed between $f_{\text{oxid}}$ and $\phi_{\text{tiller}}$.

$f_{\text{air}}$, $por_{\text{acro}}$, and $por_{\text{cato}}$: These three parameters are related to soil composition. The posterior values of $f_{\text{air}}$ increased compared to the model prior indicate a higher fraction of air in the soil. The decrease in $por_{\text{acro}}$ and $por_{\text{cato}}$ indicates decreased porosity in the acrotelm (which can contain both water or air) and catotelm (which can contain only water) respectively. $f_{\text{air}}$ and $por_{\text{acro}}$ are positively correlated, indicating more air in a more compact acrotelm environment with less water. A higher amount of air in the acrotelm can have a positive effect on diffusion. In the model, the diffusivity of $CH_4$ in air is estimated to

be four orders of magnitude larger than in water. On the other hand, lower porosity in the catotelm can reduce ebullition due to the limited porous space in the soil, which holds less water. This decreases the capacity to retain excess $CH_4$ and release it when the solubility threshold is reached. (see Kallingal et al. (2023) and Wania et al. (2010) for details).

$\lambda_{\text{root}}$: $\lambda_{\text{root}}$ played a crucial role in this optimisation. After optimisation, this parameter got a significantly lower value of 10.25 cm compared to the prior. It seems that the optimisation, when generally trying to reduce the emission from the model, has a

tendency to reduce the decay length of root biomass in the soil. The posterior parameter value closely aligns with the values reported in Kallingal et al. (2023) (10.47 cm). The optimisation results indicate a much shallower soil profile for the majority of root decay activities and $CH_4$. Given that most of the peat decomposition activities are assumed to occur in acrotelm, the reduction in the magnitude of $\lambda_{\text{root}}$ could substantially facilitate diffusion. $\lambda_{\text{root}}$ has not shown any strong correlation with any of the other parameters, but its weak negative correlation with $r_{\text{moist}}$, $r_{\text{moistan}}$, $f_{\text{oxid}}$, and $f_{\text{air}}$ suggests that soils with shallow root

depth exhibit increased moisture response and air fraction, with higher oxygen availability.

## 4.2 Posterior model estimates

Overall, the optimisation successfully reduced the model-data misfits, with some variation in the degree of improvement among different sites. This improvement is evident from the substantial magnitude change in the total cost function, RMSE and uncertainty from the prior to the posterior, and the estimated $\chi^2$ values ( see Table 5, Fig. 4, Table 6, and the Table B1 in

appendix). It should be noted that this successful improvement occurred even when assimilating data from multiple sites with diverse climatic conditions.

The approximately 95% improvement in the fit, as measured by the cost function, demonstrates the effectiveness of the GRaB-AM algorithm in sampling from high-probability regions. In general, such a large reduction in the magnitude of the final cost function value can be interpreted as the model becoming overly tuned to the specific dataset, which risks a loss

of generalisability. However, the high magnitude of the posterior cost, 79,296.4, along with the high chi-square value of 8.6, indicates a slight underfitting. A portion of this underfitting in posterior estimate can be attributed to the inability of GRaB-AM to capture the collective variability in such a large and diverse EC flux observations (see the Fig. 5). GRaB-AM assumes that the assimilated fluxes are normally distributed, however, in reality, such normal distributed fluxes usually do not occur in nature. In fact, the observed fluxes can take various forms of distributions, often with significant outliers, and multiple peaks in

the distribution. Although LPJ-GUESS is a well-developed model, there are still several processes affecting the calculation of





$CH_4$ emissions that are still not fully included in the model. For example, the assumption of zero wind speed above wetlands, the lack of detailed representation of ebullition, and the inadequate representation of wintertime emissions are some of the process in need of improvement. Another main reason for this underfitting could be the variability in the model input data. It has been observed that the model is highly sensitive to precipitation and temperature inputs. Temperature alone can explain a

large proportion of the variation in the $CH_4$ simulations (Aalto et al., 2024), which lies beyond the constraints of GRaB-AM. And even though we used the meteorological measurements at the sites, there may still be biases in the representativeness of these measurements compared to the flux footprint.

Examining the sites individually, the four sites with a significant reduction in RMSE and a high $\chi^2$ value imply enhanced accuracy in the model predictions but indicate similar underfitting as discussed before. More attention should be given to these

sites, as there may be potential for further improvement in capturing the variability of the assimilated data. On the other hand, the seven sites showing the smallest RMSE reduction and low $\chi^2$ values indicate less model improvement, but the model captures the variability in the assimilated data. The three sites with a high reduction in RMSE and low $\chi^2$ values indicate good performance in model prediction and in capturing the variability. No sites have appeared with low RMSE reduction and high $\chi^2$ values, which indicates that the predictions after optimisation have improved in at least one way or the other. No systematic

trends are observed in these matrices, which indicates that the reason a few sites show considerable improvement in both RMSE and $\chi^2$ is more due to the external factors such as model structure, and assimilated or input data rather than the assimilation framework. Even though no correlations are observed between the types of wetlands and their locations, we note that the sites that showed a considerable reduction in RMSE, such as Abi, Att, Bon, Deg, Lom, Los, Sii, Uoa, and Zar, are those that are boreal or arctic in nature, missing only Sco and Zak. Except for Los, all sites located in the temperate region, namely Hue, Ole,

Bib, and Zar, showed comparatively lower reductions in RMSE (see Table 2 and Fig. 4).

Compared to the total assimilated flux observations, the posterior estimate showed a slight underestimation of -38.1 $gCm^{-2}$ (see Sect. 3.4.1). The assimilated sites did not exhibit any consistent patterns of over or underestimation (Fig. 5). Testing the GRaB-AM algorithm at the Siikaneva site Kallingal et al. (2023) observed systematic underestimation in LPJ-GUESS posterior over many years. Employing multiple sites with varying climatic variability has proven beneficial in resolving this

issue, as sites like Bib, Deg, Sco, etc., exhibit both overestimation and underestimation in consecutive years. On the other hand, Kallingal et al. (2023) observed a considerable reduction in RMSE for Siikaneva in their single-site experiment, whereas in this study, being one among the 14 sites, the optimisation happened for Siikaneva was comparatively limited. This is rather expected in a multi-site setup, as the model must balance variability across multiple flux observations, which can degrade the fit at any individual site in favor of overall performance and generalisability.

The dominance of only one or two components after optimisation (see Fig. 7) indicates possibilities of biases. Verifying and resolving this issue might be achievable through component-wise assimilation into the model using data from all three components and local hydrology observations. However, this will be challenging due to the unavailability of data, especially of the ebullition. Measuring ebullition fluxes poses significant challenges, primarily attributed to the pronounced spatio-temporal variability. Ecosystems exhibit rapid, momentary surges in fluxes, reaching exceptionally high levels within seconds, inter-

spersed with prolonged periods of negligible ebullition (Canadell et al., 2022)





Emission of CH$_4$ from wetlands can vary significantly across latitudes, largely influenced by the growing season and climatic conditions. When assimilating data from a single site into the LPJ-GUESS model, Kallingal et al. (2023) encountered difficulties in accurately capturing winter-time emissions. We faced a similar issue, where the model emitted zero CH$_4$ when temperatures dropped below freezing point. However, observational data showed that wetlands continue to emit CH$_4$ even under sub-zero conditions (Ito et al., 2023; Treat et al., 2018; Aalto et al., 2024). This discrepancy highlights a notable limitation of the LPJ-GUESS model: its high sensitivity to temperature inputs for CH$_4$ emissions, and microbial CH$_4$ production and consumption are strongly inhibited in cold and frozen soils. The study by Ito et al. (2023), in which they compared cold-season CH$_4$ fluxes simulated by 16 models, shows that many similar models exhibit the issue with underestimating wintertime emissions. The same was observed in the study by Aalto et al. (2024), in which 6 models were compared. In both studies, LPJ-GUESS was a participant. One of the main observations by Ito et al. (2023) is that LPJ-GUESS is one of the models that discretely suppressed emissions under sub-zero temperatures and set a clear temperature threshold for CH$_4$ production. This could limit the GRaB-AM framework from adjusting the model's wintertime emissions and capturing the overall variability. Consequently, the algorithm compensates for the winter model-data mismatch by adjusting summer values. One potential model modification could involve incorporating mechanisms to simulate microbial activity under frozen conditions, snowpack insulation, and detailed representation of soil temperature dynamics, allowing for CH$_4$ emissions even when surface temperatures drop below zero. It has been observed that models representing detailed microbial activities and soil temperature profiles, such as the WETMETH model (Nzotungicimpaye et al., 2021), could simulate wintertime emissions on a comparable scale. A future study should involve using the GRaB-AM framework on such a somewhat simpler model to compare and verify its performance.

### 4.3 Validation of the posterior model and budget estimation of northern wetlands

Through selecting 14 different wetlands as described in Sect. 2.2, we aimed to equip the optimised parameters with the capability to accurately represent different types of wetlands, irrespective of their specific climatic and geographical features. Our validation analysis suggests that the optimised parameters achieved the goal to a large extent (see Sect. 3.5). In general, the optimised parameters perform better in representing different types of wetlands, especially bogs. A mismatch of 102 $gCm^{-2}$ is observed between the total flux observation and the total posterior model estimate, which is a large value compared to the result from the optimisation. Given this observation and the less constrained nature of sites like Wpt and Lgt, which are temperate, future studies might consider the necessity of different parameter sets for different wetland types. This is evident from the study of Treat et al. (2018), which shows that the behavior of wetlands can be completely different based on their type and location, especially during the non-growing season. Here it is worth noting, as mentioned in Sect. 4.2, that the majority of temperate sites used for optimisation also exhibited lower fit improvement in terms of cost function.

There is a significant shortage of reliable, long-term CH$_4$ flux observations from wetlands, which is a major challenge for studies like ours. Many of the wetland sites used in this analysis only provided measurements during summer months, or contained substantial data gaps. This limitation makes it difficult for any assimilation algorithm to fully capture the natural



variability in the data and could potentially lead to biased misinterpretations. Having used all the main sites with long-term
measurements for assimilation, it was challenging to identify additional measurement sites for validation.

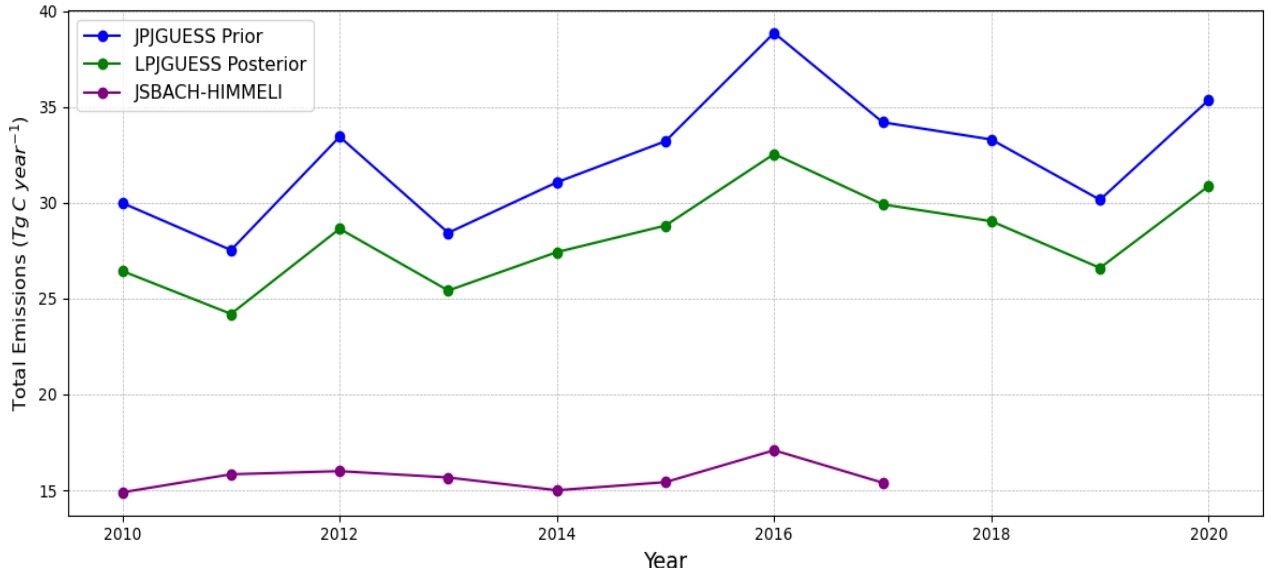

**Figure 11.** Time series of the total annual $CH_4$ estimates above 45°N. Prior and posterior $CH_4$ estimates from LPJ-GUESS, from 2010 to 2020 are shown in blue and green respectively. $CH_4$ estimate from JSBACH-H, from 2010 to 2016, is shown in purple.

Both the assimilation and validation experiments showed an underestimation in the posterior compared to the prior. The large-scale simulation above 45°N also showed a similar tendency, with around a 12.87 % reduction in the posterior compared to the prior (see Figures 10 and 11 and Sect. 3.5). We compared our results against $CH_4$ wetland emissions from JSBACH-H. This comparison primarily indicates that the change in the LPJ-GUESS simulated emissions after optimisation is in the
direction towards a better agreement with JSBACH-H fluxes, even though the comparison with JSBACH-H still suggests that LPJ-GUESS overestimates emissions. However, other studies, such as the one by Nzotungicimpaye et al. (2021), reported 24.825 Tg C $y^{-1}$ from wetlands north of 45 °N, which aligns much better with the posterior LPJ-GUESS estimate. Also, the study by Aalto et al. (2024) shows that the 16 wetland models they used simulated annual $CH_4$ emissions from wetlands of 45 °– 90 °N during 2000–2020 as 20.1 Tg C $y^{-1}$. We also compared the resulting posterior to the observations quantified
for the same domain. The gridded data product of northern wetland $CH_4$ emissions, based on upscaling EC flux observations estimated by Peltola et al. (2019), resulted in an average value of 25.98 Tg C $y^{-1}$ (23.77 - 28.2 Tg C $y^{-1}$). The results indicate that the LPJ-GUESS posterior is close to their quantified observation. However, the large uncertainty in various estimates of wetland $CH_4$ emissions makes it difficult to compare the results with other studies. For example, as mentioned in Sect. 3.5, when compared with the GCP estimate of wetland $CH_4$ emissions above 60 °N, the prior emissions from LPJ-GUESS show
better agreement. Nevertheless, the GCP estimate also reports a wide range, from 4.5 to 13.5Tg C $y^{-1}$, which would also cover the LPJ-GUESS posterior.





## 4.4 Possibilities and Limitations of GRaB-AM

The GRaB-AM algorithm incorporates the adaptation mechanism with Rao-Blackwellisation, which recursively updates the covariance of the proposal distribution to capture the dependence among different parameters. Such a recursive update of the proposal distribution can improve the efficiency of MCMC by allowing the search algorithm to take larger steps in the parameter space, while still accounting for parameter inter-dependencies. This is particularly important for high-dimensional and correlated parameter spaces. In the optimisation process laid out here, with flux observations from multiple sites and with a relatively high-dimensional parameter space of a highly non-linear model, the GRaB-AM algorithm is particularly beneficial because it enhances the exploration of a wider parameter space while adapting the proposal distribution over iterations. The algorithm has the potential to incorporate other types of natural data distributions and account for temporal correlations in the data. Based on the results of this study and the results form Kallingal et al. (2023), it is evident that the algorithm is useful for optimising complex models but can also be easily applied to simpler, more linear models such as WETMETH (see Sect. 4.2 for the discussion). Additionally, there is potential for the algorithm to be used in optimising other model variables, such as $CO_2$ fluxes, vegetation, or soil dynamics etc. where long, reliable time-series data are available.

While GRaB-AM is a powerful technique for highly non-linear models, there are several considerations and potential challenges when applying it to multi-site assimilation. (a) GRaB-AM can be computationally intensive, especially in high-dimensional parameter spaces or when dealing with a large number of sites. In this study, it took around 480 computational hours to complete 100,000 iterations on an AMD Ryzen Threadripper processor. (b) Adaptation mechanisms in GRaB-AM need to be carefully tuned to balance parameter exploration. Improper tuning may lead to suboptimal exploration of the parameter space. Additionally, the algorithm requires tuning hyperparameters that control the speed of adaptation, and its performance is sensitive to these choices, making it challenging to find appropriate hyperparameter values to begin with. (c) GRaB-AM assumes that observational data are Gaussian distributed, which is not always true when considering natural variability in multi-site flux data. The data distribution may be Lorentzian, Bernoulli, heavy-tailed, or take other forms. (d) Temporal correlations in the data are not addressed in the current form of GRaB-AM. (e) We encountered difficulties in efficiently minimising the total misfit, i.e., simultaneously reducing the misfit across multiple sites. This is due to the largely different cost function values for individual sites, caused by the varying conditions at each location. As a result, scaling was applied independently to each site to homogenise the contribution (weight) of each site in the optimisation process. However, this weighting of each site in a multi-site assimilation is a general challenge in multi-site data assimilation experiments and does not pertain to the GRaB-AM technique.

## 5 Conclusions

This study aimed to optimise the simulation of $CH_4$ emissions from natural wetlands in the LPJ-GUESS DGVM using eddy-covariance flux measurement data obtained from 14 diverse natural wetlands, characterised by variations in temporal, spatial, and/or climatic features. Ten selected model process parameters with the greatest influence on wetland $CH_4$ flux simulation are optimised using the Global Raoblackwellised Adaptive MCMC (GRaB-AM) algorithm within a Bayesian framework as



a follow-up study of Kallingal et al. (2023). Following the optimisation, the study used flux observations from five different wetlands, which again differ in their temporal, spatial and bioclimatic features, to validate the results of the optimisation. The optimisation results showed a substantial enhancement in the model's capacity to align with observed CH$_4$ fluxes, with a total reduction of approximately 50 % in RMSE and an approximately 53 % reduction in total uncertainty. The discrepancy between the modelled and observed values decreased from 1068.5 $gCm^{-2}$ to 38.1 $gCm^{-2}$. For wetlands above 45 °N, the total mean

annual emission from the posterior LPJ-GUESS estimate is 7.46 Tg C y$^{-1}$. Validation results demonstrate that four out of five sites reduced RMSE, contributing to an overall reduction of approximately 58.8 %. Given the remaining mismatches between observations and simulations and the presence of less constrained sites, future investigations will focus on individual sites, and grouping them based on their bio-geo-climatic characteristics, to examine if they need to be parameterised with different sets of parameters. Additionally, further studies are planned to quantify CH$_4$ emissions from boreal and temporal wetlands on

large spatial scales, using the optimised parameters, and to validate them against independent atmospheric observations, i.e., atmospheric CH$_4$ observations provided by the European ICOS observation network. Another intended outcome of this study is to make use of the error correlation derived from the study as prior input to the atmospheric CH$_4$ inversion model, such as Lund University Modular Inversion Algorithm (LUMIA) (Monteil and Scholze, 2021).

*Code and data availability.* The GRaB-AM code and data used for this article are available at Zenedo data deposition.The LPJ-GUESS

model code can be obtained at LPJ-GUESS. Please contact the site PIs if the site observations are intended to be used for other purposes than in this publication.

## Appendix A: Data source description

Among the sites used for assimilation, Bib (JP-BBY), Bon (US-BZB), Deg (SE-Deg), Hue (DE-Hte), Los (US-Los), Ole (US-ORv), Sco (CA-SCB), Uoa (US-Uaf), and Zar (DE-Zrk) are collected from Fluxnet datasets (the IDs in the bracket corresponds

to Fluxnet) (Fluxnet, Delwiche et al. (2021); Pastorello et al. (2020)). For the site Abi, CH$_4$ data is collected from ICOS, and the climate data is obtained from SMHI. Data for Att was collected from Ameriflux. For the site Lom (FI-Lom), climate observations of 2006 are taken from the Fluxnet site mentioned above. Observations for the remaining years are obtained from the station Principal Investigator (PI). Precipitation and temperature data for Sii are taken from FMI, and CH$_4$ data and short-wave radiation data for Sii are collected from AVAA-SMEAR (See Kallingal et al. (2023) for details). For the site Zak, the data

was taken from GEM CH$_4$ and GEM climate. The data for Che, Lgt, Sfn, and Wpt, used for validation, were collected from the Fluxnet datasets mentioned above. Climate data for Myk was obtained from SITES, and the CH$_4$ data were obtained from station PIs.



## Appendix B: Result of optimisation

**Table B1.** A comprehensive overview of un-weighted (uw) prior and posterior cost values, RMSE reduction in percentages, and the $\chi^2$ values estimated for all sites individually and together.

| Site | Prior cost value (uw) | Posterior cost value (uw) | RMSE reduc. (%) | $\chi^2$ (uw) | Site | Prior cost value (uw) | Posterior cost value (uw) | RMSE reduc. (%) | $\chi^2$ (uw) |
|------|------|------|------|------|------|------|------|------|------|
| Abi | 717913.5 | 14818.9 | 63.5 | 22.6 | Los | 163304.4 | 8711.5 | 63.42 | 11.8 |
| Att | 451259.9 | 19867.7 | 68.1 | 20.3 | Ole | 25368.6 | 2693.4 | 52.2 | 4.7 |
| Bib | 54339.3 | 1274.8 | 46.0 | 3.1 | Sco | 58147.0 | 644.4 | 36.7 | 2.0 |
| Bon | 23300.2 | 1906.2 | 70.9 | 6.8 | Sii | 58183.1 | 12406.5 | 59.2 | 16.0 |
| Deg | 10510.6 | 873.4 | 51.3 | 1.3 | Uoa | 130041.1 | 5172.5 | 68.4 | 9.2 |
| Hue | 5033.6 | 1018.4 | 27.16 | 0.96 | Zak | 3770.6 | 1357.8 | 4.3 | 2.1 |
| Lom | 54172.8 | 7672.8 | 55.9 | 9.12 | Zar | 7949.7 | 877.3 | 42.6 | 1.2 |
| | | | | | Total | 1763294.9 | 79296.42 | 50.30 | 8.6 |

## B1    Summer and winter anomaly estimation

Summer and winter anomalies of flux observations, prior, and posterior estimated separately for all 14 sites used can be seen in Fig.B1 . The figure also provides details about the years in which the model either underestimated or overestimated the emissions. It is clear that neither the simulation nor the observation follows any common seasonal patterns or trends. This indicates that $CH_4$ emissions from wetlands are generally highly dependent on the variabilities in the underlying climatic variables, and the same holds for the model. A detailed analysis of the correlation and sensitivity between the model's $CH_4$
emission and input climatic variables can be seen in Kallingal et al. (2023).





**Figure B1.** Summer and winter anomalies were estimated from the averages of the summer months (April to September) and winter months (October to March). The black, green, and purple dashed lines represent flux observations, prior, and posterior values, respectively. Dots and plus signs denote summer and winter data points of the season, respectively.

High deviations were observed in the summer anomaly compared to the winter anomaly at all the sites. In general, in the majority of cases, the model was capable enough to capture trends shown by the observational anomaly, though there were differences in magnitude. For example, for the sites Bon and Zak, the model was successful in capturing all the summer and winter trends of the observation. Notably, the high positive anomaly of Abi in 2016 and of Ole in 2015, etc., and the high negative anomaly of Att in 2015, of Deg in 2017, of Low in 2017, etc., were also captured by the model.





## B2 The annual sum estimation of CH₄

Figure B2 shows the annual sum estimation of CH$_4$ from ten out of fourteen sites used in the study. Remaining four sites are illustrated in Sect. 3.4 of the main paper. The figure illustrates that, after optimisation, most sites exhibited improved annual CH$_4$ estimation throughout the year. However, for the site Hue, the model consistently failed to capture the observation pattern
in most years, except for 2013. For the site Att, particularly for the year 2016, the model displayed shortcomings. On the other hand, Att in 2016 completely aligned with the observed value for both prior and posterior estimations.

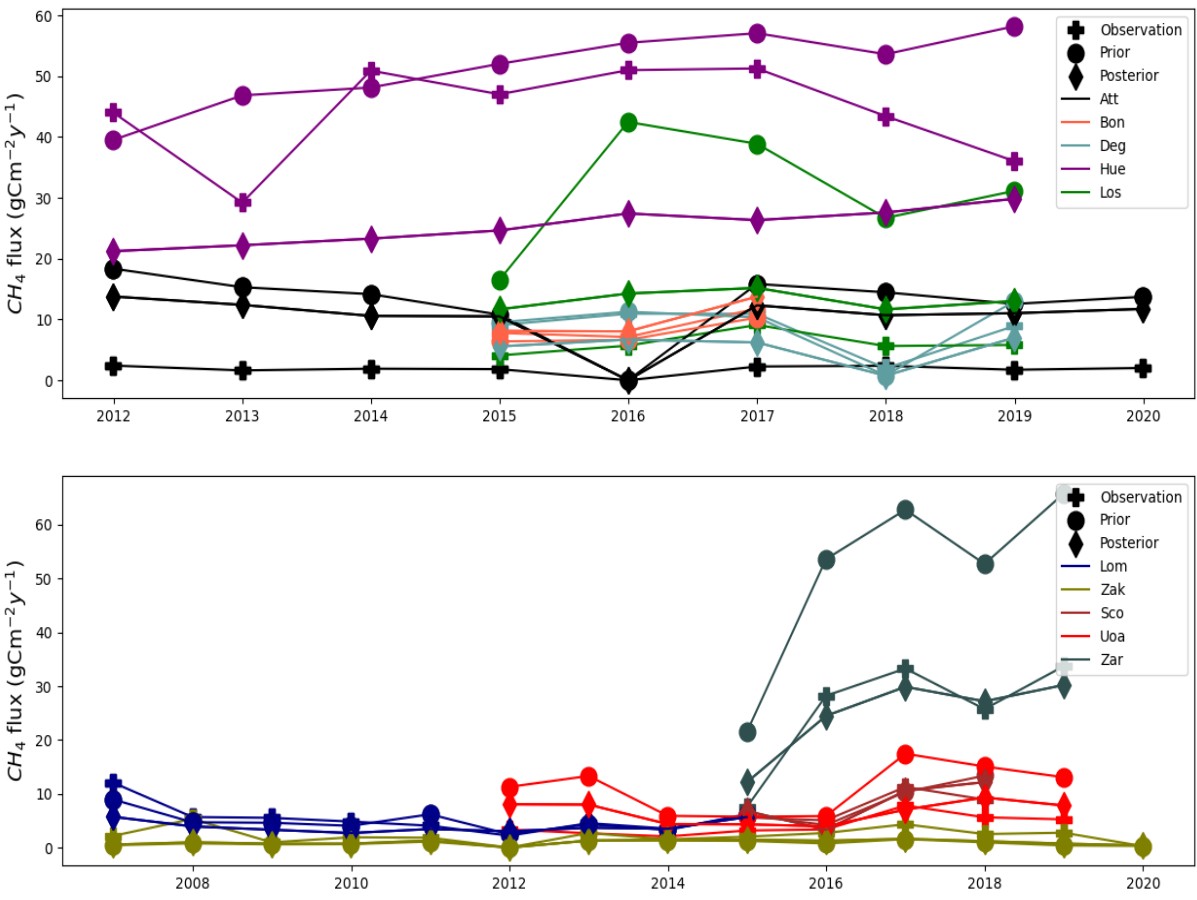

**Figure B2.** The annual sum estimation of CH$_4$ from 10 out of 14 sites used in the study. The sites are represented in different colors with distinct markings to distinguish between Observation, Prior, and Posterior.





## Appendix C: Time series estimation of validation sites

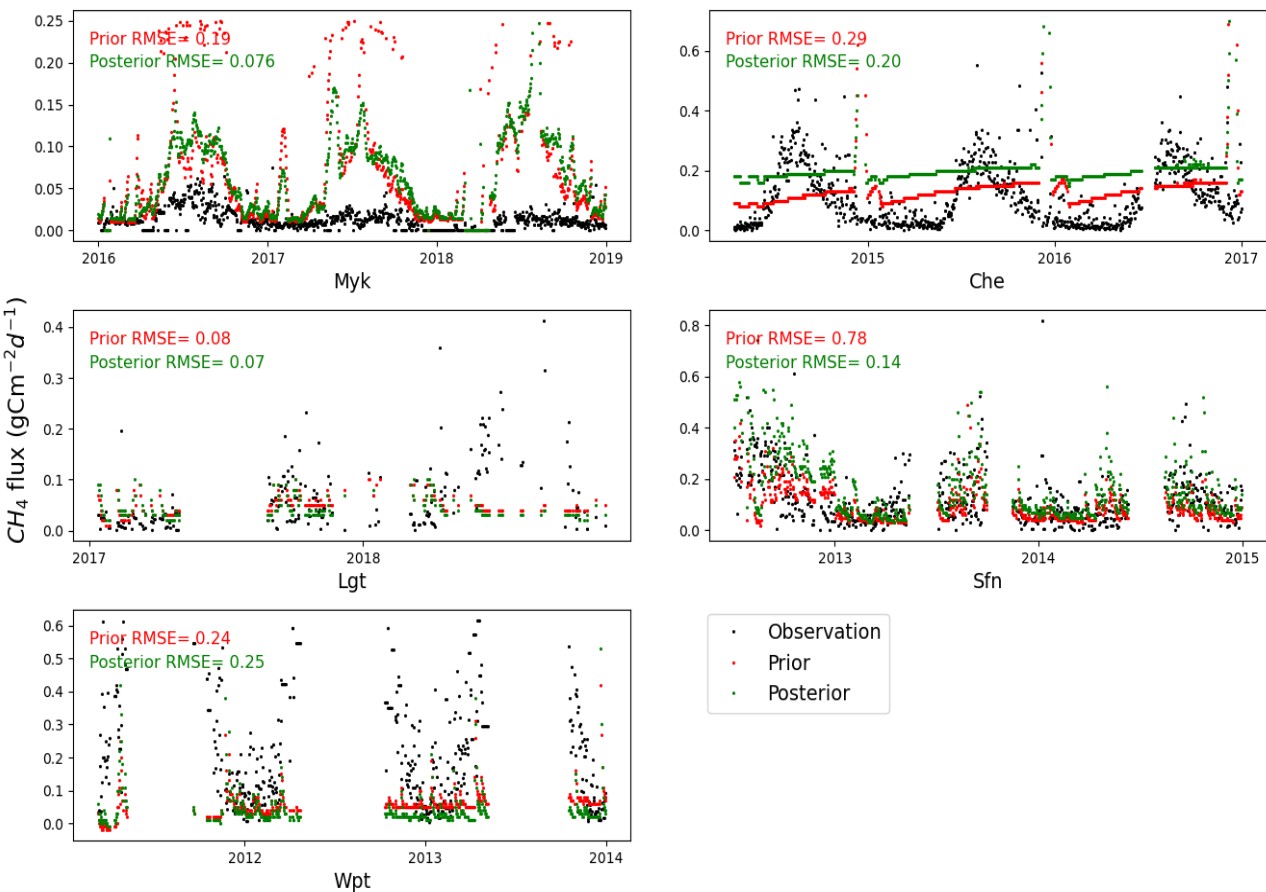

**Figure C1.** The prior and posterior CH$_4$ simulation from the LPJ-GUESS compared with flux observations from five different wetland sites used for validation. The black dots represent the CH$_4$ flux observations. The red dots depict the prior simulation using the prior model parameters and the green dots represent the posterior simulation. Three-day running averages are calculated from the original time series. In most of the figures, a few outliers on the vertical axis have been removed for better visualisation.

*Author contributions.* Conceptualisation was undertaken by JTK and MS. Methodology was formulated by JTK, JL, and MS. PM assisted in setting up the multi-site simulation in LPJ-GUESS. MA provided the CH$_4$ flux observations collected at Lompolojänkkä. PV and PW
provided the flux observations collected at Mycklemossen. Setting up the GRaB-AM and writing the original draft was carried out by JTK. Editing were performed by JTK, MS, JL, PM, JR, PV, PW, and MA. All authors have read and agreed to the published version of the manuscript.





*Competing interests.* The authors declare that they have no conflict of interest.

*Acknowledgements.* We would like to express our gratitude to Ivan Mammarella, Annalea Lohila, Kirsty Langley, Jutta Holst, and Elin
Humphreys for their guidance on data repositories. Special thanks to Sadat Ismayil for assisting with computational resources. Additionally,
we acknowledge the Fluxnet database, the Ameriflux database, the Integrated Carbon Observation System (ICOS), Greenland Ecosystem
Monitoring (GEM), the Swedish Meteorological and Hydrological Institute (SMHI), the Swedish Infrastructure for Ecosystem Science
(SITES), the Institute for Atmospheric and Earth System Research (SMEAR-INAR) at the University of Helsinki, and the Finnish Meteoro-
logical Institute (FMI) for providing open access to their data.

*Financial support*: This research has been supported by the Strategic Research Area: Biodiversity and Ecosystem services in a Changing
Climate (BECC), Lund University, and is a contribution to the Strategic Research Area: ModElling the Regional and Global Earth system
(MERGE). BECC and MERGE are funded by the Swedish government.



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
