# Peer review of "Assimilating Multi-site Eddy-Covariance Data to Calibrate the CH4 Wetland Emission Module in a Terrestrial Ecosystem Model"

_EGUsphere, 2024_

## Author Response (AR1)

**Reply letter to the Referee #1**

*We are very thankful to the Referee for the constructive comments. In the following, we have addressed the Referee's comments. The manuscript has also been revised accordingly.*

*Note: The reviewer comments are referred to in blue font type throughout the texts, and the authors' responses are referred to as black font.*

**General comments:**

This manuscript presents a follow-up study to a recent paper (Kallingal et al., 2023) which showed that model parameters for the LPJ-GUESS model could be tuned using a Markov Chain Monte Carlo type approach. While Kallingal et al only tuned model parameters based on data from one site, this manuscript uses data from 14 different sites to assess whether the parameter fitting approach works on a broader scale. Study results show that the approach does indeed work, and authors use the resulting model parameters in LPJ-GUESS to estimate total mean annual CH4 emissions above 45N. The manuscript topic is timely and valuable, and improving global estimates of wetland methane emissions is crucial to reducing uncertainty surrounding this critical greenhouse gas. The authors have clearly put significant time and effort into this work, and I appreciate how exhaustively they have presented their findings. **However, in my opinion the manuscript would benefit significantly from some work to reduce it's length. As is the paper is 29 pages, which feels much too long for the scope of work and results presented. I suggest authors spend some time trimming down the text as I think this would greatly improve the reader's ability to follow the work. Some of the components could be moved to the appendix, and some may be best to eliminate (see specific comments on section 2.6 and 3.4.1).**

We appreciate the reviewer's positive feedback and constructive suggestion. We acknowledge that the manuscript became longer than initially planned, partly due to additional text and sections added during the previous revision process. In response to your recommendation, we have now moved Sections 2.6, Section 3.4.1and Table 5 to the appendix to streamline the main text and improve overall readability.

Line 22-23 – Please elaborate on what the difficulties have been, this will better justify the need for your work.

The difficulties in estimating wetland $CH_4$ emissions over large landscapes, as discussed in Saunois et al. (2020), arise from several key factors related to both the nature of wetlands and the limitations of available measurement methods. These include:

- **Spatial heterogeneity** of wetlands, as their structure, vegetation, and hydrology vary significantly across short distances;

- **Temporal variability**, since $CH_4$ emissions fluctuate strongly with changing environmental conditions such as temperature, water table depth, vegetation phenology, and seasonal dynamics;

- **Scale mismatch**, because eddy-covariance (EC) flux measurements are made at site-level and represent a limited footprint, making it difficult to extrapolate findings to larger regions without introducing substantial uncertainty.

Acknowledging the reviewer's suggestion, we have revised the sentence accordingly to better highlight these challenges as :

*While current in-situ measurement techniques such as eddy-covariance (EC) flux observations are promising for drawing assumptions at local scales, studies to date have faced difficulties in estimating wetland $CH_4$ emissions over large landscapes, primarily due to the spatial heterogeneity of wetlands, strong temporal variability in emissions, and the limited coverage of long-term flux observations. Line 21-24.*

Line 26-28. Again, I suggest adding more detail on what model complexities have been added over time to give the reader more context.

We thank the reviewer for the suggestion. In response, we have revised the text in the manuscript to provide more detail on the model developments over time. The updated text now reads:

*These models were simple in structure, and later, more attention was given to improving process representation, through the studies of mainly Segers and Leffelaar (2001), Gedney et al. (2004), and Zhuang et al. (2006). In the past decade, more advanced models have incorporated additional complexities such as dynamic water table fluctuations, temperature-dependent microbial activity, substrate diffusion, and oxidation in the soil column, along with broader applications across diverse wetland types and climatic conditions (e.g., Wania et al., 2010; Ringeval et al., 2010; Susiluoto et al., 2018). Line 29-31*

Lines 60-62. Is it unclear if you are focusing on just high-latitude wetlands, or are also including temperate wetlands – please clarify.

We thank the reviewer for the comment. However, we would like to note that the focus on **arctic, boreal, and temperate wetlands** has been specified in several places throughout the manuscript, both before and after Line 60. For example, this is clearly stated in the abstract (Lines 8–13), as well as in Lines 61–62, and further reiterated in Section 2.2 and Tables 2 and 4.

Line 111 – What is your justification for using this approach? The standard of care within the eddy covariance flux community is more advanced than this, please see Vuichard and Papale et al., 2015 (doi:10.5194/essd-7-157-2015) for details and update your analysis accordingly.

Thank you for pointing this out. We acknowledge that the standard approaches within the eddy covariance community, such as the gap-filling methods outlined by Vuichard and Papale (2015), provide more robust estimates, particularly for flux variables. However, in this case, the variable is climate *forcing data*, which are used as model inputs. Our choice to back-fill short gaps (<14 days) was made to preserve time series continuity with out much efforts. Since these gaps were infrequent and short in duration, we believe the impact on model performance is minimal. This way of gap-filling—commonly known as forward fill or last observation carried forward—is widely used in environmental modelling and time series analysis to maintain data continuity during short, infrequent gaps.

Section 2.2 – I find this section greatly lacking in necessary detail. At no point do authors even clarify what type of flux measurements they are using.

We thank the reviewer for the comment. We would like to clarify that the study uses **eddy-covariance (EC) flux measurements of $CH_4$**, which are explicitly stated in several places in the manuscript. This is mentioned in the **title**, the **abstract (Lines 2–6)**, the **introduction (Line 22–23 and Line ~45)**, and again in **Section 2.3**. However, to avoid any ambiguity, we have now added a

clarifying sentence to **Section 2.2** explicitly stating that the study uses EC CH$_4$ flux data for all selected sites as:

*"In total we have used 18,437 data points of daily EC CH$_4$ flux observations for assimilation,"*

Furthermore, authors do not say whether they acquired their data through established data-sharing portals (such as those from AmeriFlux or ICOS), or if data were provided directly to them via site PIs. If data are not acquired from the regional networks, which have established protocols for data QAQC, authors must specify how data were cleaned and processed. Assuming data are from regional networks, authors should provide DOI links to the data sets (in addition to the already-provided links to papers describing those data) as well as the office 5-character site names to help readers understand where their data are coming from. If authors aren't using the standardized, gap-filled data from the FLUXNET-CH4 product, please explain why.

We thank the reviewer for these helpful suggestions. The details regarding data collection, including web addresses and associated references, are already provided in Appendix A (Data source description) and mentioned in the main paper. Most of the CH$_4$ flux datasets were obtained directly from site PIs or publicly accessible research repositories, as indicated in Appendix A .

We used the **standard gap-filled meteorological (climate) data from FLUXNET**, as they are essential for driving the LPJ-GUESS model consistently across all sites. However, we **intentionally did not use gap-filled CH$_4$ flux data** from the FLUXNET-CH$_4$ product or other sources. Instead, we retained the observed CH$_4$ flux time series with their original data gaps. This decision was made to avoid introducing potential artefacts or bias associated with imputation or gap-filling methods, and to ensure that only directly observed data were assimilated. The cost function was calculated using valid (non-gap) observations only.

Section 2.6 – A lot of these presented functions are standard definitions and do not need to be written out, or at least could be saved for an appendix.

We thank the reviewer for this helpful suggestion. These standard function definitions were originally added during the first revision. However, we agree that they do not need to be included in the main text. In response, we have now moved Section 2.6 to the appendix (Appendix B: Statistical metrics) to improve the readability and focus of the manuscript.

Table 3 – It would be helpful to report the # of site-years of data, rather than # of data points available. Furthermore, please specify range of data (year 1 – year x)

Thanks for your observation. However, we would like to clarify that, the **number of site-years** and the **range of years** (e.g., *Year 1 – Year X*) are already reported in **Table 2**. That said, we believe that the **number of observations** is also an important metric to report—especially in the context of **data assimilation**, where the underlying assumption is that a larger number of observations can lead to more robust and reliable results.

Moreover, the **number of observations** is directly used in the calculation of the **chi-squared ($\chi^2$)** values, which are central to our evaluation of model performance

Table 5 – Prior values from table 5 do not match Table 1, please fix this. Values in Table 5 also do not match black dashed "Prior mean" lines in Figure 2. Furthermore, Table 5 does not contain any information that is not graphically shown in Figure 2, so should be moved to an appendix.

We thank the reviewer for this detailed observation. The prior values shown in Table 1 refer to the **model priors,** i.e., the default parameter values used in LPJ-GUESS to define the central points of the parameter ranges. These were used to construct the prior distributions and set the bounds for the MCMC sampling.

In contrast, the "prior values" in Table 5 refer to the **initial random values** (station priors) used to start the MCMC chains. These are sampled from the parameter prior distributions and represent the first state of the chain, not the default model values. We have now clarified this distinction in the caption of Table 5. Table 5 provides additional information about the prior and posterior cost values, as well as the aforementioned station priors. Please note, however, that Table 5 has now been moved to the appendix.

Figure 3. – most of these correlations are not significant, please only include correlation numbers for significant relationships.

While we agree that statistical significance is useful for interpretation, the intent of Figure 3 is not solely to establish statistically significant relationships, but also to examine potential patterns, parameter interactions, and structural dependencies in the posterior distribution. These insights are particularly valuable for identifying issues such as equifinality or parameter redundancy challenges that may not be fully captured by statistical significance alone, especially in high-dimensional posterior spaces.

The correlations presented here are based on the full set of posterior samples generated through MCMC, which reflect the joint uncertainty structure of the parameters. In this context, the magnitude and direction of the correlations—even when not statistically significant—can still provide important diagnostic information about parameter identifiability and sensitivity. While we appreciate the reviewer's suggestion, we have chosen to retain all correlation values in Figure 3 to preserve completeness and transparency in the presentation of the posterior structure.

Figure 5 – Please change the coloring so it is readable by people with color blindness.

Thanks for pointing this. The color is now changed in to Teal – Dark Purple – Black.

Figure 6a – Instead of using different marker shapes to delineate different lines, I suggest instead using different line dash patterns as this will be easier to distinguish when points overlap.

We appreciate the reviewer's suggestion. We agree the issue with overlapping markers, which was mainly due to the figure's vertical orientation—specifically, the height was mistakenly set a bit large in the original version. We have now corrected the figure dimensions, which improves the spacing and visibility of the markers. We have retained the original marker shapes, as they allow consistent visual reference across related figures throughout the manuscript.

Section 3.4.1 I question whether you should include a detailed assessment of how the estimated transport pathways have shifted in importance between prior and post. First of all, flux data are not able to distinguish between CH4 emitted from these various pathways, thus there are no data to check the validity of these results. Second, you acknowledge in line 442 that LPJ-GUESS has a "lack of detailed representation of ebullition", thus further throwing into question the results presented in section 3.4.1. Without additional work to check the veracity of these estimates, I suggest eliminating this section.

We thank the reviewer for this thoughtful comment and agree that $CH_4$ flux measurements do not distinguish between emission pathways, and that the current LPJ-GUESS representation of processes such as ebullition is relatively simple. However, we believe that the transport component

analysis still provides useful interpretive value particularly in revealing how the optimisation process adjusts parameter sensitivities across sites and how the model structure interacts with observational constraints.

These variations and shifts generally indicate the high sensitivity of specific parameters to different emission components and expose gaps in the current process representations. For example, if the model fails to establish vegetation at a site due to unfavourable hydrological or temperature conditions, plant-mediated transport may be suppressed—this likely occurred at Hue and Att, where plant-mediated emissions were minimal in both the prior and posterior results. Furthermore, the optimisation algorithm may shift the relative importance of transport components to compensate for variability in the flux data, particularly in the absence of wintertime emission processes in the model. These are structural aspects that are difficult to diagnose and even harder to explain. Still, they are valuable for understanding model behaviour under uncertainty.

We fully acknowledge the limitations of this analysis, particularly the lack of independent observations for validating the transport pathways. To reflect this, and in response to the reviewer's concern, we have moved the content of Section 3.4.1 to the appendix (Sec.B1).

Line 443 – citations for these model deficiencies?

Thanks for pointing this. We now have sited: Kallingal, Jalisha T., et al. "Optimising CH 4 simulations from the LPJ-GUESS model v4. 1 using an adaptive Markov chain Monte Carlo algorithm." *Geoscientific Model Development* 17.6 (2024): 2299-2324.

Line 437-439: This seems like a major shortcoming, how do you justify using a methodology that requires data normality with data that may be far from normal?

We agree with the reviewer that the assumption of Gaussian-distributed fluxes is a simplification, and that $CH_4$ EC observations often exhibit skewness, heavy tails, and outliers. This is a well known limitation in many data assimilation frameworks, especially those based on classical Bayesian approaches, including the one used in this study. While it is in principle possible to assume alternative, non-Gaussian distributions that better reflect the empirical distribution of multi-site EC flux data, doing so introduces significant complexity into the assimilation framework. The primary objective of this study, however, is to evaluate the GRaB-AM algorithm in a multi-site context.

The following details have now been added to the paper to justify the use of a methodology that requires data normality (See Lines: 396-403): *GRaB-AM assumes normally distributed fluxes, but in reality, observed fluxes often deviate from normality, exhibiting outliers, skewness, and heavy-tails. However, the simplification to normality is a direct result of applying the commonly used quadratic loss function in data assimilation, and aids tractability of the MCMC-implementation. Introducing more complex error distributions would require estimation of the parameters of those distribution, significantly increasing both the number of parameters to estimate and the computational expense. Such an expansion, while possible and interesting future work, is outside the scope of this study..*

**Reply letter to the Referee #2**

We are very thankful to the Referee for the constructive comments. In the following, we have addressed the Referee's comments. The manuscript has also been revised accordingly.

*Note: The reviewer comments are referred to in blue font type throughout the texts, and the authors' responses are referred to as black font.*

**General comments:**

The manuscript by Kallingal et al. covers an interesting and timely topic appropriate for Biogeosciences. The study focuses on using data assimilation techniques to improve CH4 flux simulations with the process-based model LPJ-GUESS. This approach addresses a much-needed yet less explored area within the CH4 research community, making the study both innovative and highly relevant. The manuscript is generally well-organized, with most parts clearly written. I have a few minor comments as follows:

**Assumption on normal distribution. The assumption that assimilated fluxes follow a normal distribution needs justification. Is this a robust assumption? How does it affect the model estimation? There are some studies suggesting the flux measurements does not follow a Gaussian distribution.**

We thank the reviewer for raising this important point. It is true that flux measurements do not necessarily follow a Gaussian distribution. However, the assumption of normally distributed fluxes follows from the common use of minmizing quadratic errors when optimizing models. For MCMC-based data assimilation frameworks we require not only a loss function but also the corresponding distribution induced by the loss function, given these restrictions the Gaussian distribution is preferable due to its computational efficiency and tractability.

The Gaussian error distribution is particularly suitable when fluxes are influenced by the aggregate effect of multiple small, independent processes, or constitute averages over many measurements (such as our aggregation from half-hourly to daily means)—as suggested by the Central Limit Theorem. Nevertheless, this assumption may not hold in all cases, especially when fluxes are shaped by extreme events, non-linear dynamics, or exhibit skewness and heavy tails.

We acknowledge that using a Gaussian likelihood can introduce biases in parameter estimation if the underlying flux distribution is substantially non-Gaussian, although this effect is typically relatively small. A bigger concern is that it may lead to under- or overestimation of parameter uncertainty, and can limit the sampler's ability to explore the full posterior distribution.

Alternative statistical approaches—such as using log-normal, gamma, or skewed-t distribution can better account for skewness, heteroscedasticity, and heavy tails in EC flux observations. However, implementing these models introduces computational challenges, including slower mixing, additional hyper-parameters and higher rejection rates in the sampling process. These issues are particularly difficult to manage in large-scale, site-level applications involving complex process-based models.

In this study, our aim was to demonstrate the application of the GRaB-AM framework (Jalisha et al., 2024) across multiple wetland types using a consistent and computationally feasible assimilation setup. We recognise the limitations of the Gaussian assumption and are currently conducting a follow-up study to extend the GRaB-AM framework by incorporating skewed and

heavy-tailed likelihood formulations that better reflect the statistical characteristics of EC CH$_4$ flux data.

**Performance visualization:** Figure 9 does not clearly convey performance improvements post-assimilation. Including a scatterplot to better illustrate this would be helpful.

A scatterplot is used to replace the old one, comparing prior and posterior RMSE values for each site. Each point represents one of the study sites. The 1:1 dashed line indicates no change in RMSE. Points below the line indicate improved model performance after assimilation.

[Figure]

**Observation uncertainty:** How is uncertainty in the observations considered in the data assimilation? Here, the daily mean values derived from half-hourly measurements are used, which the temporal coverage of half-hourly measurements can affect the derived daily mean value.

Thank you for the insightful comment. We acknowledge this limitation. The CH$_4$ data used in our study were collected from station PIs or publicly available datasets, including FLUXNET, where we did not have complete information over the data collecting or processing procedures.

Specifically, when averaging half-hourly observations to derive daily means, the variance should ideally scale as 1/n, where $n$ is the number of available observations per day. Therefore, days with full coverage (e.g., 48 half-hourly measurements) would have a smaller uncertainty compared to days with fewer observations (e.g., only 5 measurements), potentially leading to large differences in variance across days. To minimise this difference, In this study we have removed days with less than 50% of half-hourly data availability.

Also due to these limitations, the uncertainty in observations is modelled using an error distribution, assuming Gaussian. This accounts for measurement noise, sensor inaccuracies, and representativeness errors. The uncertainty in the daily mean flux estimates is propagated into the assimilation framework using an error covariance matrix which has explained in the paper from line 149 to 158 as below:

*Considering the difficulty of calculating error correlations in the flux observations, we only considered errors in individual observations, i.e., we did not consider off-diagonal elements in specifying the observational error covariance matrix R in the cost function (see Eq. 2 in the main paper). Estimating the exact observation error for each site is again challenging. Assigning a constant percentage error for all measured values could introduce a bias, as it would result in high error values for measurements with high magnitudes and very low error values for observations with small magnitudes. To overcome this challenge, we followed the procedure introduced by Knorr and Kattge (2005) for the case of assimilating CO2 eddy covariance observations and assign a threshold value set at 5% of the variance of the distributions of observations, calculated separately for each site. Values below this threshold are identified, and a uniform error is assigned to them (see Table 3 in the main paper) . An error of 30% is estimated for the observations greater than the threshold values.*

**Annual budget calculation:** Does the PEATMAP dataset cover all wetlands or only peatlands? If it is restricted to peatlands, the exclusion of other wetland types (e.g., mineral wetlands) makes comparisons with the GCP results problematic. This needs clarification.

We thank the reviewer for raising this important point. The PEATMAP dataset specifically focuses on peatlands. The dataset is the result of a meta-analysis of geospatial information regarding peatlands, which are a type of wetland. We agree that this restricts our total $CH_4$ emission estimates to a subset of wetland types. However, we note that the GCP $CH_4$ budget ensemble also incorporates **a similar peatland-specific dataset (Hugelius et al., 2020)** (line 863-868) for high-latitude regions (primarily above 60°N), where peatlands dominate. This was integrated alongside the broader WAD2Mv2.0 product in order to improve spatial accuracy in these regions.

**Insights by wetland types:** Presenting results by sites is useful but limited in scope. Can the authors share insights categorized by wetland types?

We appreciate the reviewer's suggestion. While we agree that presenting results by wetland types (e.g., bogs, fens, marshes) could offer an additional layer of insight, this would require extensive restructuring of the analysis and figures, which goes beyond the current scope. However, we would like to point out that information regarding wetland types and climate zones is already provided throughout the manuscript. Specifically:

- **Table 2** clearly categorizes the 14 assimilation sites by **wetland type** (e.g., bog, fen, marsh) and **climate zone** (arctic, boreal, temperate) .

- In **Section 2.2**, we mention that the sites were selected to represent a diversity of wetland types including **fens, mires, bogs, marsh, and wet tundra** .

- In **Section 4.2**, we reflect on the apparent trend that **arctic and boreal wetlands** showed greater improvements in model performance compared to **temperate sites** .

- Additionally, **Section 4.3** discusses how the model performance varies across **different wetland types**, and highlights potential limitations in how the model currently represents them (e.g., challenges with winter fluxes in cold climates or suppressed emissions in some marshes) .

Given that these aspects are already integrated across the paper, and that the current presentation aligns with the study's main aim of testing a generalisable assimilation framework across diverse sites, we have opted not to restructure the results around wetland type at this stage. However, we acknowledge the value of this perspective and will consider this stratification in future work, where

additional site-specific analyses and larger sample sizes can support more robust group-wise interpretations.

**Specific comments:**

**Line 280:** The term "contribution" is unclear. What specific aspect or variable does this refer to?

Thanks for the comment. This has now changed as:

*"Contribution of CH4  from among….."*

**Line 307:** The dominance of only two components after optimization is interesting. Are these findings supported by observations? What are the implications of this result?

We thank the reviewer for this insightful question. The emergence of dominance by only two $CH_4$ transport components—typically **diffusion and plant-mediated transport**—in the posterior estimates reflects the way the optimisation algorithm reallocates emission pathways to better match the observed total fluxes under the structural constraints of the model.

While direct observational partitioning of $CH_4$ emissions by transport pathway is limited, studies have shown that **diffusion and plant-mediated transport** often represent the **major $CH_4$ pathways in wetlands**, particularly during the growing season (Wania et. al 2010). **Ebullition**, while important in certain conditions (e.g., shallow, warm, or peat-rich systems), is known to be highly **episodic and spatially variable**, which can reduce its contribution to daily average fluxes as used in this study.

The **lack of detailed process representation for ebullition in LPJ-GUESS** (as noted in Section 4.4) may also limit its expression in the model, particularly during optimisation. Furthermore, the model cannot rely on external drivers or process-based triggers (e.g., pressure buildup, water level fluctuations) that typically govern ebullition, possibly resulting in its lower weight in the posterior estimates.

The implication of this finding is twofold:

1. It suggests that the **current model structure**, combined with the assimilation of total flux observations, **favors smoother and more continuous pathways** (diffusion and plant-mediated transport), which are more consistent with the temporal resolution of the EC flux data.

2. It also highlights the need for improved representation and validation of **individual transport processes**—particularly ebullition—within ecosystem models, possibly supported by targeted field observations and component-wise partitioning studies in future work.

However, we have decided to move the section to the appendix based on the suggestion of Reviewer #1.

**Line 511:** Is "underestimation" the correct word?

I have changed it with the word reduction: *Both the assimilation and validation experiments showed a reduction in the posterior compared to the prior.*